

**Basin-scale multi-objective simulation-optimization modeling for**
**conjunctive use of surface water and groundwater in northwest China**
Jian Song[a], Yun Yang[b], Xiaomin Sun[c], Jin Lin[c], Ming Wu[d], Jianfeng Wu[a,*], Jichun Wu[a]
[a] Key Laboratory of Surficial Geochemistry, Ministry of Education; Department of
Hydrosciences, School of Earth Sciences and Engineering, Nanjing University, Nanjing,
210023, China
[b] School of Earth Sciences and Engineering, Hohai University, Nanjing, 210098, China
[c] Nanjing Hydraulic Research Institute, National Key Laboratory of Water Resources and
Hydraulic Engineering, Nanjing, 210029, China
[d] Institute of Groundwater and Earth Sciences, Jinan University, Guangzhou, 510632, China
[*]Corresponding author: Jianfeng Wu (jfwu@nju.edu.cn; jfwu.nju@gmail.com)



**ABSTRACT**

20       In the arid inland basin of China, the long-term unregulated agriculture irrigation from

surface water diversion and groundwater abstraction has caused unsustainability of water
resources and degradation of ecosystems. This requires integrated management and
conjunctive use of surface water (SW) and groundwater (GW) at basin scale to achieve
scientific decision supports for water resources allocation in China. This study developed a
novel multi-objective simulation-optimization (S-O) framework for sustainably conjunctive
use of SW and GW in Yanqi Basin (YB), a typical arid region with intensive agricultural
irrigation in northwest China. The S-O model integrates the new epsilon multi-objective
memetic algorithm ($\varepsilon$-MOMA) with the MODFLOW-NWT based simulation model for
examining the hydraulic interactions between SW and GW. Four conjunctive management
objectives, involving maximizations of total water supply rate, groundwater storage change and
surface runoff inflow to Bosten Lake, and minimization of total water delivery cost, were
considered to explore the tradeoffs between socioeconomic development and environmental
demands. The combined multi-objective SW and GW management model can achieve the
tradeoffs in high-order objective spaces by considering groundwater abstraction in the
irrigation districts and surface water diversion from the river, so as to avoid the prevalence of
decision bias caused by the low-dimensional optimization formulation. Decision-makers are
then able to identify their desired water use schemes with preferred objectives in the
post-optimization and achieve maximal socioeconomic and ecological benefits. Furthermore,
three representative runoff scenarios under changing climatic conditions were specified to
quantify the influence of decreasing runoff in Kaidu River on the YB water management.
Results show that runoff reduction would be of great negative impact on the total water supply,
surface runoff inflow to the lake and regional groundwater storage in the aquifer. Therefore, the
integrated SW and GW management is of critical importance for the protection of the fragile



hydro-ecosystem under changing climatic conditions.
**Keywords**: Multi-objective optimization; water resources management; conjunctive use; Yanqi
Basin; Bosten Lake
**1. Introduction**

In arid and semi-arid inland basin, the intensive irrigation for agricultural development

caused the deterioration of natural ecosystem sustained with scarce water resources (Wichelns
and Oster, 2006; Wu et al., 2016). In general, the irrigation water is diverted from groundwater
(GW) abstraction and surface water (SW) diversion in the densely populated oasis regions in
northwest China (Liu et al., 2010; Wu et al., 2014). Therefore, the conjunctive management of
GW and SW is essential for the requirement of local economic development and
eco-environmental conservation (Khare et al., 2006; Safavi and Esmikhani, 2013; Singh, 2014;
Hassanzadeh, et al., 2014; Wu et al., 2016). Yanqi Basin (YB) is a typical oasis in an arid
inland basin located to the southern Tianshan Mountains in Xinjiang Province, northwest China.
The surface water resource in YB is mainly composed of a river and a lake, namely Kaidu
River and Bosten Lake, the biggest freshwater inland lake in China (Wang et al., 2014; Zhou et
al., 2015). Kaidu River supplies approximately 95% of total inflow to Bosten Lake (Gao and
Yao, 2005; Liu et al., 2013; Yao et al., 2018) which is the major water source of the Kongqi
River recharged by an artificial pumping station built in 1983. Therefore, the water supply
scheme in YB dominates the water balance in Bosten Lake and has a significant influence on
the Kongqi River and the lower reaches of Tarim River where the serious water crisis has taken
place. With the intensive agricultural development, surface water diverted from Kaidu River
can no longer meet crop water requirements. Thus, groundwater became the alternative water
source for crop production whereas the excessive groundwater exploitation has caused the
deterioration of local ecosystem associated with the decline of groundwater level and altered
the hydraulic interaction between GW and SW (Hu et al., 2007; Zhang et al., 2014; Tian, et al.,


2015, Yao et al., 2015). For this reason, the integrated SW and GW management is essential for
rational utilization of water resources in the arid inland basin due to the physical water scarcity.

In the water resources planning and management, the simulation-optimization (S-O)

methods can provide optimal schemes to guide and inform stakeholders (Maier et al., 2014).
Evolutionary algorithms have been integrated with simulation model to tackle intricate SW and
GW management model due to the effectiveness of solving non-linear and multimodal
optimization problems (McPhee and Yeh, 2004; Yang, et al., 2009; Safavi and Esmikhani, 2013;
Singh and Panda, 2013; Rothman and Mays, 2013; Wu et al., 2014; Parsapour-Moghaddam et
al., 2015; Wu et al., 2016). Yang et al. (2009) considered conflicting bi-objectives with the
conjunctive use of GW and SW to achieve optimal pumping and recharge schemes. Rothman
and Mays (2013) developed an optimization model including cost control, aquifer protection
and growth objectives using multi-objective genetic algorithm. Wu et al. (2016) performed the
temporal optimization of monthly volume of surface water diverted from Heihe River by
linking a physical-based integrated modeling with a simple single-objective management
model. However, these studies rarely consider many-objective optimization in the basin-scale
water management with conjunctive use of SW and GW. The management model including the
typical single objective or bi-objective formulation probably results in the decision bias (*i.e.*,
cognitive myopia or short-sightedness) due to the sub-optimal solution only considering the
fewer preference criteria (Kasprzyk et al., 2012, 2015; Woodruff et al., 2013; Matteo et al.,
2019). Therefore, for water management with the strong interactions between SW and GW in
the basin-scale water cycle, the optimal water use practice calls for decision-maker to consider
multiple conflicting management objectives. In general, the management objectives are
composed of maximizing water use demands (*e.g.*, total volume of surface water and
groundwater use), minimizing the capital and operation costs of water delivery and minimizing
the effect of water resources exploitation on the hydro-ecosystem to ensure sufficient





environmental flow (*e.g.*, the regional groundwater storage and surface runoff inflow to the
terminal lake).

Multi-objective evolutionary algorithms (MOEAs) can obtain the tradeoff solutions that

cater to multiple competing objectives and reflect comprehensive decision information for
practitioners in real-world applications (Beh et al., 2017; Eker and Kwakkel, 2018). However,
many-objective optimization often suffers from the domination resistance phenomenon
(Purshouse and Fleming, 2007; Hadka and Reed, 2013), which shows that the diminishing
Pareto-sorting capacity triggers many non-dominated solutions in the population and then
results in stagnation of evolutionary search. Hadka and Reed (2013) developed a novel Brog
MOEA, which employed auto-adaptive six recombination operators, $\varepsilon$-box technique for the
Pareto sorting and injection strategy to avoid stagnation of evolutionary search and archived
optimal solutions in addressing many-objective optimization. In order to enhance the local
optimality of solutions, a memetic algorithm composed of the biological process of natural
selection and cultural evolution capable of local refinement was applied to ensure the
convergence of the MOEA (Sindhya et al., 2011; 2013). This study attempts to utilize the
$\varepsilon$-dominance concept, the modified auto-adaptive recombination operators to alleviate
domination resistance problem, and a local search operator to enhance the local optimality of
archived solutions with the framework of NSGA-II (Deb et al., 2002). The improved algorithm,
named epsilon multi-objective memetic algorithm ($\varepsilon$-MOMA), is applied to the many-objective
optimization of conjunctive management of SW and GW for agricultural irrigation in YB.

In this study, a regional numerical model using MODFLOW-NWT (Niswonger, 2011) is

developed for quantitatively evaluating water budget and interaction of river-lake-groundwater
in YB. The model is calibrated according to long-term series of observation data during
simulation period from 2003 to 2013. Kaidu River and Bosten Lake are simulated with
Streamflow-Routing package (SFR2) (Richard and David, 2010) and Lake package (LAK3)





(Michael and Leonard, 2000). The lake and river simulation is calibrated based on observed
lake level and runoff data at the gaging stations, respectively. Then, a well-calibrated model is
linked with the novel $\varepsilon$-MOMA to explore optimal water supply schemes which consider
multi-stakeholders' benefits simultaneously. Moreover, in order to encourage decision-makers
to use the optimized schemes, an interactive tool is employed to visualize and analyze all the
Pareto-optimal solutions and provide suggestions on the practical operation of water allocation.
Kaidu River mainly gains water from seasonal precipitation that runs off the mountainous
landscape and snow and glacier that melts in the upper Tianshan Mountains region known as a
main water tower in the Central Asia. Dashankou gauging station is the dividing point between
midstream and downstream of Kaidu River and the inlet of the basin where most of the river
runoff flows into YB. Therefore, the runoff variation in Dashankou station, which is highly
sensitive to the changes of precipitation and glacier mass loss dominated by the climate change,
greatly affects the water resources and water cycle in Kaidu River watershed. Three
representative runoff scenarios under the future climatic conditions are then specified to
explore the effects of runoff reduction in Kaidu River on the integrated SW and GW
management practices.

This study firstly constructs a multi-objective SW and GW management model to

consider water supply and environmental benefits including regional groundwater storage and
surface runoff inflow to lake. Then the spatial conjunctive optimization of surface water
diversion and groundwater abstraction is implemented using the novel multi-objective
evolutionary algorithm ($\varepsilon$-MOMA). The optimization results demonstrate that water managers
can achieve the optimal schemes constrained by satisfying the water demands and sustaining
the fragile hydro-ecosystem in YB. The implications from the optimization under the runoff
reduction scenarios also provide valuable insights for water use practices in the face of climate
changes in the arid inland basin.



## 2. Methodology


As shown in Fig. 1, this study aims to develop a multi-objective decision-making
framework to optimize irrigation schemes of surface water diversion and groundwater
abstraction for the integrated SW and GW management. The optimal schemes can assist
decision-makers to achieve water demands and ensure water balance of hydro-ecosystem in
YB. The optimization framework includes three main modules and their details are stated in
the following sections.
**Figure 1.**
*2.1 Problem formulation*
Module I in the optimization framework is to formulate an integrated SW and GW
management model to implement water resources management in the basin. The water
utilization patterns for irrigation are composed of diverting surface water from the inland reach
of river basin and pumping groundwater from the regional aquifer. Therefore, the decision
variables consist of the volume of surface water diversion in the aqueduct system and
groundwater abstraction in the irrigation districts. In general, the optimal water supply
strategies are maximizing the total volume of water supply and minimizing the capital and
operation costs of water delivery. However, in the arid inland basin with water scarcity, the
intensive agricultural development requires enough irrigation water to ensure local economic
development while the sustainability of hydro-ecosystem also needs to follow specific
requirements for maintaining environmental flows. For example, the excessive surface water
diversion can significantly reduce the runoff inflow to the terminal lake, which causes obvious
decline of lake level and results in the degradation of local hydro-ecosystem associated with
the lake. Meanwhile, immoderate exploitation of groundwater stored in the aquifer to offset the
surface water shortage triggers a series of environment problems (*e.g.*, dramatic decrease of


groundwater storage). Therefore, decision-makers should consider the total water supply rate
and the cost of water delivery from multiple sources as socioeconomic metrics, and describe
the runoff inflow to lake and groundwater storage as environmental metrics. Then, water
managers can assess water use practices by weighing the four preference criteria. The
performances of all schemes are evaluated based on the well-calibrated numerical model. The
detailed formulation of management model can be seen in Section 3.3. Finally, the optimization
model formulates water use practices as decision variables, socioeconomic and environmental
metrics as management objectives, practical limitation of water exploitation and water
demands for hydro-ecosystem as constrained conditions for the basin-scale SW and GW
management.
*2.2 Optimization process*

Module II in the optimization framework illustrates the algorithmic process of $\varepsilon$-MOMA.

The metaheuristic algorithms are superior to the classical optimization methods and have been
successfully applied to water resources management and planning (Maier et al., 2014) due to
the ability to solve complex problems with nonlinear, nonconvex and high-dimensionality
features. To address domination resistance phenomenon in the many-objective optimization,
the proposed algorithm integrates a $\varepsilon$-box technique, adaptive multi-operators recombination
and a local search operator into the framework of NSGA-II. The key techniques can be
recapitulated as follows.

The $\varepsilon$-box technique proposed by Laumanns et al. (2002) attempts to ensure convergence

and diversity of Pareto-optimal solutions. Moreover, decision-makers can define the minimum
resolution of objective vector with epsilon vector to satisfy their acceptable precision target and
restrict the archive size. This study implemented the $\varepsilon$-dominance archive process after the fast
non-dominated sorting of offspring individuals and alleviated the difficulties derived from the



192 domination resistance.

193   The auto-adaptive multi-operator recombination proposed by Hadka and Reed (2013) is a

194 promising technique to select optimal operator for various optimization problems. The

195 crossover probabilities of each operator are updated periodically based on the proportion of the

196 solutions generated by each operator in the $\varepsilon$-dominance archive. The recombination strategy is

197 essential for the complex many-objective and real-world optimization due to the inability to

198 know a prior the optimal recombination operator. This study integrated the multiple

199 recombination operators (*i.e.*, simulated binary crossover (SBX), differential evolution (DE),

200 simplex crossover (SPX), parent-centric crossover (PCX), Laplace crossover (LX), uniform

201 mutation (UM)) into the $\varepsilon$-MOMA to enhance search ability in higher order objective spaces.

202   The archived solutions are operated based on Gaussian perturbation in the neighborhood

203 of the evolutionary individuals. Given an archived individual $\mathbf{v}=(v_1,v_2,v_3,\ldots,v_n)$, the mutated

204 individuals can be stated as:

205   $$\mathbf{v}^+=\left(v_1,v_2,\ldots,v_i+p\times\left(m_i-w_i\right),\ldots,v_n\right) \tag{1}$$

206   $$\mathbf{v}^-=\left(v_1,v_2,\ldots,v_i-p\times\left(m_i-w_i\right),\ldots,v_n\right) \tag{2}$$

207 where $\mathbf{v}=(v_1,v_2,\ldots,v_n)$ is an *n*-dimensional decision variable vector; $\boldsymbol{m}=(m_1,m_2,\ldots,m_n)$ and

208 $\boldsymbol{w}=(w_1,w_2,\ldots,w_n)$ are two individuals randomly selected from the archive; $c$ follows standard

209 Gaussian distribution. The process is effective with the probability of $1/n$ (Chen et al., 2015).

210 The $\varepsilon$-MOMA revives the local search operator in every several generations. Therefore,

211 $\varepsilon$-MOMA possesses the ability of highly effective global search with adaptive recombination

212 operator and epsilon domination to find higher quality and diverse solutions with local search

213 operator in solving intricate many-objective optimization problem.

214 *2.3 Visual analytics of Pareto-front*

215   In the many-objective optimization, it is difficult for water managers to distinguish the





performance of single solution and discover desired schemes without the detailed visual
analytics. Module III used an interactive visual analytics package, DiscoveryDV (Hadka et al.,
2015; Kollat and Reed, 2007), to explore and analyze water management practices in the
high-order objective spaces. The package employed multi-dimensional coordinate plot and
parallel coordinate plot (Inselberg, 2009) to visualize Pareto-optimal solutions. Visualizing
performance objectives can assist stakeholders to compare with the scheme before the
optimization and select key tradeoff schemes with a clearer perspective (Matteo et al., 2019;
Maier et al., 2014). Moreover, decision-makers can eliminate redundant schemes based on the
preferred objectives or concerns and filter the optimal subsets those probably adopted by the
experienced practitioners.
**3   Case study**
*3.1 Study area*
YB is a typical oasis in an arid inland desert basin in the southern Tianshan Mountains,
Xinjiang Province, northwest China and includes Yanqi County, Hejing County, Bohu County
and Heshuo County, with a total area of about 7600 km$^2$ (Fig. 2). In the model domain, the
northwest is mountainous and the south is a low-lying desert, and the terrain slopes from
northwest to lower southeast. YB is located in the temperate zone of continental desert climate
with an annual mean temperature of 14.6 °C, an annual precipitation of 50.7-79.9 mm, and a
potential evaporation of 2000.5-2449.7 mm (Mamat et al., 2014). The basin is mainly
composed of the Kaidu River, Huangshuigou River and Qingshui River. Kaidu River originates
from the Hargat Valley and the Jacsta Valley in the middle part of the Tianshan Mountain with
a maximum altitude of 5000 m and ends in Bosten Lake (Xu et al., 2016). Kaidu River is the
largest inland river in YB which provides annual mean runoff of $3.41 \times 10^9$ m$^3$ (Wang et al.,
2013) and plays an utmost role in protecting the lake and its surrounding ecology and





environment. The Dashankou station is the dividing point that divides the mainstream of the
river into middle and lower reaches. In YB, the runoff in Kaidu River is mainly diverted for
agricultural irrigation and finally flows into Bosten Lake, which contributes to about 95% of
the water recharge for the lake (Yao et al., 2018). Bosten Lake is a largest freshwater inland
lake in China covering the area of about 1005 km$^2$ with a length of 55 km and a width of 25 km.
The lake water volume is approximately $8.8\times10^9$ m$^3$, with an average depth of 7 m and a
maximum depth of 17 m (Xiao et al., 2010). The evaporation and an artificial discharge by a
pumping station built in 1983 control the outflow of the lake. As shown in Fig. 2, the pumping
channel starting from the outflow point is utilized to divert the lake water to recharge Kongqi
River and supply water to the lower Tarim River. The dam is built to sustain higher lake level
for the water diversion. Therefore, Bosten Lake is a main water source to the lower reaches of
Tarim River, which had suffered from severe degradation of ecological environment resulted
from unregulated water exploitation in the past few decades. The Chinese government
implemented the Ecological Water Conveyance Project in 2000 to sustain ecosystem in the
lower Tarim River by transferring water from Bosten Lake (Xu et al., 2007; Hao and Li, 2014).
However, YB is an intensive agricultural area where is mostly made up of farmland growing
crops of tomato and pepper. The irrigation water demands accounted for 90% of the total water
consumption in the basin due to the rapid increase of farmland area in the recent years (Yao, et
al., 2018). Consequently, scientific water management strategies should strike for balancing the
demands for existing irrigation and eco-environmental water use to sustain enough water
inflowing from Kaidu River to the lake and the aquifer.

This study selects the core part of YB comprising the majority of irrigation districts.

Kaidu River plays a vital role in regulating and maintaining regional water balance in YB. The
modeled domain (Fig. 2) is bounded by the mountains on the northwest, the Huangshuigou
River on the northeast, swamp areas and Bosten Lake on the south. As shown in Fig. 2, an





aqueduct system conveys and redistributes the surface runoff from the mainstream of Kaidu
River and the fully penetrating wells are used to pump groundwater from the aquifer.
**Figure 2.**
*3.2 Numerical model*
The numerical model in this study is modified from the previous work of Wu et al. (2018)
using MODFLOW-NWT and then performed a multi-objective optimization based on the
corrected model. The specified boundary conditions in the model are illustrated in Fig. 3. The
northwest border was defined as the flow boundary to simulate recharge of groundwater runoff
in the interface between mountains and plain. Huangshuigou River and southwest border were
considered as the specified head boundary based on observed groundwater level. The swamps
and Bosten Lake were modelled using the general head boundary (GHB) package and LAK3
package, respectively. The bathymetric contours of Bosten Lake were used to confirm the lake
bottom topography. Kaidu River and aqueducts were simulated using the SFR2 package. The
simulation period in the transient model was defined from November in 2003 to October in
2013. Totally 20 stress periods were discretized, two periods for each year including
non-irrigation period (from November to next March) and irrigation period (from April to
October of each year), in the entire simulation period. The key model parameters for both SW
and GW were adjusted to reproduce the fluctuation of groundwater levels at the observation
wells and streamflow in the gaging stations (*i.e.*, Yanqi and Baolangsumu stations) as shown in
Fig. 3. The observed lake levels in the simulation period were employed to calibrate the model.
**Figure 3.**
The model calibration was manually implemented by the trial-and-error method. The
Nash-Sutcliffe Efficiency (NSE) was applied to evaluate the simulated precision of runoff and
lake level. The predicted accuracy of groundwater head was assessed based on root mean





square error (RMSE) and correlation coefficient ($R$). The performance criteria can be stated as:

$$\text{NSE} = 1 - \frac{\sum_{t=1}^{T}\left(y_{m,t} - y_{o,t}\right)^2}{\sum_{t=1}^{T}\left(y_{o,t} - \overline{y}_o\right)^2} \tag{3}$$

$$\text{RMSE} = \sqrt{\sum_{i=1}^{N}\left(y_{m,i} - y_{o,i}\right)/N} \tag{4}$$

$$R = \frac{\sum_{i=1}^{N}\left(y_{m,i} - \overline{y}_m\right)\left(y_{o,i} - \overline{y}_o\right)}{\sqrt{\sum_{i=1}^{N}\left(y_{m,i} - \overline{y}_m\right)^2 \times \sum_{i=1}^{N}\left(y_{o,i} - \overline{y}_o\right)^2}} \tag{5}$$

where $y_{m,t}$ and $y_{o,t}$ are the simulated and observed runoff or lake level for $t$th stress period,
respectively; $T$ is the number of stress periods; $y_{m,i}$ and $y_{o,i}$ are the simulated and observed
groundwater head at the $i$th observation well, respectively; $N$ is the number of observation
wells; $\overline{y}_m$ and $\overline{y}_o$ are the average value of simulated and observed data. Fig. 4a and 4b
compare the simulated and observed runoff at Yanqi and Baolangsumu Stations for the periods
between 2004 and 2012 and suggest that the long-term fluctuation of runoff in Kaidu River can
be well reproduced by the model. Fig. 4d shows the simulated groundwater heads have a
good-fit with observed heads at the all observation wells with RMSE of 1.8 m and R of 0.98.
Fig. 4e compares the observed and calibrated groundwater level over time in the three
observation wells and the groundwater variation trend in the irrigation and non-irrigation
period can be achieved.
**Figure 4.**
The interaction between Bosten Lake and the aquifer is dominated by the hydraulic
conductivity of the lakebed, of which value is very small owing to the existence of the thick
low-permeability sediment in the region. The main inflow term of the lake is the surface runoff
from Kaidu River which has been calibrated based on the runoff in the gauging stations. The





recharge for the lake from precipitation is not significant in the arid inland basin. The outflow
terms are mainly composed of the evaporation and artificial pumping to divert water from the
lake to Kongqi River. The local water resources authority in YB provided the data of artificial
pumping in the simulation period. However, the average evaporation in Bosten Lake calculated
using potential evaporation data or Penman's equation is not accurate because the temperature
and relative humility exhibit the significant difference over the approximately 945.0 km$^2$
evaporation surface. Therefore, the observed lake stages were applied to calibrate evaporation
rate in the lake. Fig. 4c illustrates the calibration results of lake level and indicates that the
decline trend of lake level can be adequately captured. Then, the water balance of Bosten Lake
can be achieved as shown in Fig. 5. In the simulation period from 2004 to 2013, surface runoff
inflow in Kaidu River represents 97.4% of the total annual inflow to the Bosten Lake. The total
annual outflow of the lake consists of 54.9% of lake evaporation and 44.2% of artificial
pumping. Therefore, the surface runoff in Kaidu River is a crucial factor to maintain the water
balance of Bosten Lake. The surface runoff inflow can be considered as a significant
performance metric to evaluate the water management practices in YB. Finally, the numerical
model has been well-calibrated and can be employed to integrated SW and GW management.
**Figure 5.**
*3.3 Management model*
The integrated SW and GW management focuses on not only the water resources
exploitation subject to social and economic benefits but also the effect of water exploitation on
environment benefits. The study formulated an integrated SW and GW optimization problem
including four management objectives: (1) to maximize total water supply rate ($f_{TWS}$); (2) to
minimize total cost of water delivery from water intake points to water use destinations ($f_{TCOST}$);
(3) to maximize the groundwater storage change of saturated zone between the beginning and





end of management period ($f_{GSC}$) which is negative when the storage decreases and vice versa;
and (4) to maximize surface runoff inflow from Kaidu River to Bosten Lake ($f_{SRI}$). $f_{TWS}$ and
$f_{TCOST}$ are defined as the metrics to satisfy the local irrigation water demands while maintain the
lower costs of water use. $f_{GSC}$ is formulated as the metric indicating the extent of groundwater
exploitation and a greater value shows a preferred situation. $f_{SRI}$ is defined to evaluate the
influence of surface runoff from Kaidu River on the water balance in Bosten Lake, which
contributes about 97.4% of the total inflow (Fig. 5). As shown in Fig. 6, the decision variables
are the total volume of surface water diverted in the mainstream of Kaidu River in the
diversion point (DP1-DP7) and groundwater abstraction in the irrigation districts (ID1-ID11).
The formulations of management model are given as follows:
$$\text{Max} \quad f_{TWS} = \sum_{i=1}^{N_p} Q_{g,i} + \sum_{i=1}^{N_d} Q_{s,i} \tag{6}$$
$$\text{Min} \quad f_{TCOST} = \sum_{k=1}^{N_t} \sum_{i=1}^{N_w} q_{g,i,k} C_g \left( H_i - h_{i,k} \right) T_k + \sum_{k=1}^{N_t} \sum_{i=1}^{N_d} q_{s,i,k} C_s T_k \tag{7}$$
$$\text{Max} \quad f_{GSC} = \sum_{j=1}^{N_g} \left( h_{end,j} - h_{ini,j} \right) Sy_j A_j \tag{8}$$
$$\text{Max} \quad f_{SRI} = f_{gaging} (\mathbf{X}) \tag{9}$$
$$\mathbf{X} = \left( Q_{g,1}, Q_{g,2}, \dots, Q_{g,N_p}; Q_{s,1}, Q_{s,2}, \dots, Q_{s,N_d} \right) \tag{10}$$
where $Q_{g,i}$ is total groundwater abstraction rate at $i$th irrigation district (m³/yr); $Q_{s,i}$ is total
volume of surface water diverted from $i$th diversion point (m³/yr); $N_p$ is the number of
irrigation districts; $N_d$ is the number of diversion point based on the locations of aqueducts; $N_t$
is the number of stress period including irrigation and non-irrigation period; $N_w$ is total number
of pumping wells; $q_{g,i,k}$ is the pumping rate at the $i$th well in $k$th stress period (m³/d); $C_g$ is the
cost per unit pumping rate per length of hydraulic lift in case of wells (0.015 CNY/m³/m); $H_i$ is
the surface elevation at the $i$th pumping well (m); $h_{i,k}$ is the groundwater level at the $i$th well in





$k$th stress period (m); $T_k$ is the length of the $k$th stress period (d); $q_{s,i,k}$ is the surface water
diversion rate at the $i$th diversion point in $k$th stress period (m$^3$/d); $C_s$ is the cost per unit
diversion volume (0.055 CNY/m$^3$); $N_g$ is the total number of active cell in the modelling
domain; $h_{end,j}$, $h_{ini,j}$ is the groundwater level at the end and beginning of management period
(m); $Sy_j$ is the specific yield at $j$th active cell; $A_j$ is the area of $j$th grid cell (m$^2$); $f_{gaging}$ outputs
the surface runoff in Kaidu River at the inflow point of Bosten Lake; **X** is a water use scheme.
**Figure 6.**
The management model consists of a set of constraints given by:
$$Q_{g,min} \leq Q_{g,i} \leq Q_{g,max} \quad Q_{s,min} \leq Q_{s,i} \leq Q_{s,max} \tag{11}$$
$$d_{max} \leq d_c \quad h_{lake} \geq h_c \tag{12}$$
$$\sum_{i=1}^{N_p} Q_{g,i} \geq TP_{min} \quad \sum_{i=1}^{N_d} Q_{s,i} \geq TD_{min} \tag{13}$$
$$Q_{out,i} > 0.0 \tag{14}$$
where $Q_{g,min}$ and $Q_{g,max}$ are the capacity of total groundwater abstraction at specified irrigation
district and $Q_{g,min}$ is uniformly assumed to $1\times10^6$ m$^3$/yr and $Q_{g,max}$ is $1\times10^8$ m$^3$/yr; $Q_{s,min}$ and
$Q_{s,max}$ are the constraints of surface water diversion at diversion point, $Q_{s,min}$ is $1\times10^7$ m$^3$/yr at
diversion points DP1 and DP2 and $5\times10^6$ m$^3$/yr at DP3-DP7, $Q_{s,max}$ is $4\times10^8$ m$^3$/yr at DP1 and
$2\times10^8$ m$^3$/yr at DP2 and $1\times10^8$ m$^3$/yr at DP3-DP7; $d_{max}$ is the maximum drawdown and must
less than the permission value $d_c$ which is set to 5 m based on the existing management
schemes; $h_{lake}$ is lake level and must greater than minimum level $h_c$ (1045 m in this study) to
divert lake water to recharge Kongqi River; $TP_{min}$ and $TD_{min}$ is the prescribed minimum water
demands of total groundwater abstraction and total surface diversion to satisfy the agricultural
development and are set to $3.0\times10^8$ m$^3$/yr and $5.5\times10^8$ m$^3$/yr based on the reports from the
local water resources authority; $Q_{out,i}$ represents outflow of the end reach of $i$th stream segment



and must greater than zeros which means the potential diversion at each diversion point does
not exceed the available streamflow in the current segment to avoid significant error of water
budgets in the optimization (Wu et al., 2015). This study aims at optimizing spatial distribution
of groundwater abstraction at different irrigation district and surface water diversion at each
diversion point. The management period was set to one year with duplicated model inputs and
parameters from November 2012 to October 2013 including the non-irrigation and irrigation
periods. Then the conjunctive management of SW and GW is implemented based on the
multi-objective optimization framework carried out in MATLAB software
(http://www.mathworks.com/products/matlab).
**4   Results and discussion**
*4.1  Pareto-optimal solutions*
This study applied $\varepsilon$-MOMA to solve the integrated SW and GW management model with
four objectives ($f_{TWS}$, $f_{TCOST}$, $f_{GSC}$ and $f_{SRI}$) to search for optimal water use schemes. The
algorithm parameters and objective epsilon values are summarized in Table 1. Fig. 7 shows a
global view of tradeoff surface in a 4-dimensional coordinate plot. The management model
consists of maximizing the $f_{TWS}$, $f_{GSC}$ and $f_{SRI}$ objectives and minimizing the $f_{TCOST}$ objective.
The $f_{TWS}$, $f_{SRI}$ and $f_{GSC}$ are plotted on the *x*, *y* and *z* axes and $f_{TCOST}$ is represented with color in
Fig. 7. The green arrow indicates the direction of optimality in each objective. It can be
observed that the trade-off relationship exists between $f_{TWS}$ and other objectives ($f_{TCOST}$, $f_{GSC}$
and $f_{SRI}$). Augmenting the total amount of water supply increases the cost of transporting water
with the solutions marked in red color and reduces surface runoff inflow to the lake and
groundwater storage at the end of management period. Therefore, the regional water resources
exploitation conflicts with the socioeconomic and environmental benefits in YB. The scheme
before optimization is marked in red square box in Fig. 7. We can see that the scheme is





located above the tradeoff surface and exhibits larger cost value. Thus, the current management
scheme is sub-optimal and can be regulated to obtain optimal performances.
**Table 1.**
**Figure 7.**
To explain the discrepancy of the Pareto approximate set, the parallel coordinates plot is
used to illustrate the tradeoff surface while the total pumping rate ($f_{TPR}$) and total surface water
diversion rate ($f_{TDR}$) are added to elucidate the effect of conjunctive use of SW and GW. In Fig.
8, the segments with higher $f_{TWS}$ exist for higher $f_{TCOST}$ and lower $f_{GSC}$ and $f_{SRI}$, indicating that
increasing water demands requires more financial investment and depletes more surface runoff
inflow to the lake and groundwater storage. The findings are consistent with the previous
inferences in Fig. 7. Moreover, the many slope segments exist between $f_{TPR}$ and $f_{GSC}$, $f_{TDR}$ and
$f_{SRI}$, which indicates that enlarging groundwater abstraction and surface water diversion are the
dominated factors for the depletion of groundwater storage and surface runoff recharge for the
lake, respectively. It is noteworthy that the variation trend of $f_{TPR}$ is very close to the change of
$f_{TWS}$ while the change in $f_{TDR}$ exists obvious difference. The increment of $f_{TPR}$ can be reached to
$4.16 \times 10^8$ m$^3$/yr whereas the growth of $f_{TDR}$ only is $1.14 \times 10^8$ m$^3$/yr across all the Pareto
solutions. Therefore, groundwater abstraction can be adjusted largely to satisfy management
objectives based decision-makers' preference whereas surface water diversion should be
restricted. The reasons behind this bias are that surface water diversion is highly sensitive to
the lake level and the intensive groundwater abstraction augments the river leakage that
indirectly causes the decrease of the available runoff.
**Figure 8.**
*4.2 Optimized management schedule*
The superiority in many-objective optimization is the full exploration of optimal solutions



to avoid the decision bias derived from the lower dimensional objective formulation. The
decision-makers can firstly analyze the performance of the Pareto solutions in the sub-problem
(*e.g.*, single or two-objective optimization) and then explore the tradeoff solutions using the
previous analysis in the higher order objective space to satisfy the multi-stakeholders' benefits.
Figs. 9a-9c illustrate the projection of four-objective Pareto solutions onto two-objective space
with non-dominated front of the sub-problem constructed by the $f_{TWS}$ and other objective
($f_{TCOST}$, $f_{GSC}$ and $f_{SRI}$), respectively. As shown in Figs. 9a-9c, Solutions 1-3 are the compromise
solutions in the non-dominated front in the two-objective sub-problem which may be selected
by the decision-makers with no preference in the certain objectives. However, these
high-performance solutions in the two-objective optimization exhibit worse performance in the
other objective spaces. As illustrated in the plots (Fig. 9), Solutions 2 and 3 have higher $f_{TCOST}$
than Solution 1 in Fig. 9a, Solutions 1 and 3 have lower $f_{GSC}$ than Solution 2 in Fig. 9b and
Solutions 1 and 2 show lower $f_{SRI}$ than Solution 3 in Fig. 9c. Therefore, the decision-makers
need identify the true compromise solution that performs well in the multiple objectives
simultaneously. In this study, Solution 4 is closest to the corresponding objective values of the
compromise solutions (Solutions 1-3) at the same time and can be a true compromise solution
in the 4-dimensional tradeoff surface. Additionally, Solution 5 has the largest objective value of
total water supply rate in the Pareto approximate set which meets constraints of maximum
groundwater drawdown and minimum lake level. Solution 6 corresponds to the compromise
solution in the non-dominated front of $f_{GSC}$ and $f_{SRI}$ which indicates the perfect performance in
the protection of regional groundwater storage and water balance of the lake.
**Figure 9.**

In this study, Solutions 4, 5 and 6 are selected to elucidate the variation of groundwater

abstraction and surface water diversion compared with the scheme before optimization
(Solution 7). The objective values of selected solutions are listed in Table 2. It can be observed



that Solution 4 can achieve similar total water supply rate while the cost of water delivery can
reduce 34.4% compared with Solution 7. The result shows that Solution 7 is sub-optimal from
the aspect of expenditure of water supply. Moreover, the surface runoff inflow to lake in
Solution 4 achieves the increment of $3.82\times10^7$ m$^3$/yr and the depletion in groundwater storage
obtains the reduction of $1.99\times10^7$ m$^3$/yr. However, $f_{GSC}$ of Solution 4 is still less than zero
which demonstrates the loss of groundwater storage compared with initial state. Therefore,
Solution 6 is a preferred water use scheme from the aspects of the maximization of
groundwater storage and surface runoff inflow to lake simultaneously. The objectives of
Solution 6 in Table 2 show reducing $1.43\times10^8$ m$^3$/yr of $f_{TWS}$ in the scheme before optimization
can achieve the increment of groundwater storage with $2.19\times10^7$ m$^3$/yr and augment $6.30\times10^7$
m$^3$/yr of surface runoff inflow to lake. Solution 5 represents the potential of water resources
exploitation in YB and can augment 26% of total water supply rate compared with Solution 7.
Interestingly, it can be found that, in Solutions 5 and 7, groundwater storage depletion
($8.39\times10^7$ m$^3$/yr) is more rapid than the reduction of surface runoff inflow to the lake ($1.85\times10^7$
m$^3$/yr), which indicates groundwater abstraction is probably preferred option to provide the
resiliency of water supply in the face of the increased water demands.
**Table 2.**
Fig. 10 illustrated the spatial distribution of the pumping rates of the selected solutions at
11 irrigation districts. As shown in Figs. 10a and 10b, Solution 4 shows groundwater
abstraction in the ID3, ID5 and ID7-ID11 can be increased in comparison to Solution 7. It can
be noted that the pumping rates in ID7 and ID9 can be largely elevated due to lower
exploitation in the past and shallow groundwater depth. The groundwater abstraction in ID1,
ID2, ID4 and ID6 should be reduced especially for the pumping rate in ID6 which exhibits
abrupt decline. As shown in Fig. 10c, Solution 5 with the maximization of $f_{TWS}$ demonstrates
that a large amount of groundwater can be abstracted in the ID5-ID9 (greater than $8\times10^7$ m$^3$/yr)





which implies water managers can implement groundwater abstraction in those districts to
satisfy the augmentation of water supply. In Fig. 10d, Solution 6 is a desired scheme with the
maximization of environment benefits in groundwater storage and runoff recharge to the lake.
The spatial differentiation of groundwater abstraction in Solution 6 is similar with those in the
4-dimensional compromise solution (Solution 4). However, Solution 6 based the pumping rates
in the ID5 and ID8 show obvious decline, which implies that water managers can lower the
groundwater abstraction in these regions to achieve more environment benefit in groundwater
storage.
**Figure 10.**
Fig. 11 illustrates the spatial patterns of surface water diversion along the main stream of
Kaidu River. As show in Fig. 11a, seven diversion points (DP1-DP7) with the reduction of
runoff are clearly identified. The runoff at the 35 km from DP1 exhibits obvious rise due to the
inflow in the tributary. The river runoff at the lake inflow point is the surface runoff inflow to
the lake that is $f_{SRI}$ objective. It can be observed that the surface runoff in the scheme before
optimization (Solution 7) in DP1 shows the abrupt decline than Pareto-optimal solutions
(Solutions 4, 5 and 6) which responds to the distribution of surface diversion in Fig. 11b.
Moreover, Solution 7 has the lowest runoff between DP1 and DP4 even though exists slight
increase in the lake inflow point. Therefore, a significant increase of surface water diversion in
DP1 controls the available runoff in the downstream segments. The water managers should
reduce the surface water diversion in DP1 to ensure sufficient runoff in the lower reaches of
Kaidu River for the adjustment of multi-stakeholders' benefits. Solution 4 is a compromise
scheme that exhibits lower runoff compared with Solution 6 from DP4 to the end of river, due
to the larger water diversion in DP4, which triggers the reduction of surface runoff inflow to
lake. Solution 5 is a potential of regional water resources exploitation in YB and has smaller
available runoff than Solutions 4 and 6 which implement more water diversion in Kaidu River.


Fig. 11c further demonstrates the interaction of surface water and groundwater along the
mainstream of the river. The upper segment (Segment I) is a losing segment that means surface
water exchange from stream to aquifer and the middle segment (Segment II) is a gaining
segment that indicates groundwater exchange from aquifer to stream. Then the lower segment
(Segment III) turns into a losing segment. It can be noted that Segment I and Segment II have
strong interaction between SW and GW whereas Segment III exhibits exchange with a lower
leakage rate. As illustrated in Fig. 11d, the distribution of total river leakage shows Solution 5
maximizing water supply corresponds to the maximum loss of runoff which is in fact caused
by the substantial groundwater abstraction and the exchange from Solutions 6 and 7 shows the
less river leakage. Consequently, groundwater abstraction is a dominated factor for the
interaction of SW and GW for the YB water management. The river leakage in Solution 4 is
obviously larger than Solution 7 which is seemingly undesired for water managers. However,
augmenting groundwater abstraction ($1.31\times10^8$ m$^3$/yr) at the cost of river leakage ($0.30\times10^8$
m$^3$/yr) can lower surface water diversion ($0.67\times10^8$ m$^3$/yr) directly from the river that is highly
sensitive to the runoff inflow to Bosten Lake. Therefore, groundwater abstraction is probably a
desired water use pattern in YB.
**Figure 11.**
*4.3 Impacts of runoff change*
Kaidu River plays a crucial role to sustain regional water balance in YB and flows through
Dashankou station (Fig. 2) into the basin. The river supplies the majorities of surface water
diversion by an aqueduct system for agricultural irrigation and constitutes about 95% of total
annual inflow to the Bosten Lake. The runoff in Kaidu River is mainly originated from
mountainous precipitation and melting glacier water in the Tianshan Mountains region.
However, the remarkable climate changes have caused a significant increase in both



temperature and precipitation over the past 50 years in Xinjiang (Li et al., 2013). The changing
climate probably increased the glacier melt and snowmelt in the upper part of Kaidu River and
then caused the growth of the river runoff between 1999 and 2002, with the highest runoff in
2002 of 5.7 billion $m^3$/year (Zhou et al., 2015). However, the long-term climate change may
reduce runoff in Kaidu River attributing to the depletion of small or mid-size glaciers and snow
line receding in the middle Tianshan Mountains region. Li et al., (2012) observed that surface
area of snow in the Kaidu River Basin reduced largely between 2000 and 2010. Therefore, it is
essential to consider the impact of runoff reduction in Kaidu River on the regional water
resources management for the local socioeconomic and environmental development.

This study implemented multi-objective optimization by resetting the runoff inflow at the

first diversion point (DP1) in Kaidu River with the duplicated model parameters and the inputs
of source and sink terms. We defined three scenarios which are to maintain the current runoff
(Scenario A0), reduce 10% of the runoff (Scenario A1) and reduce 20% of the runoff (Scenario
A2), respectively. In the management model, the constraint of lake level is altered to the
smaller value (1044.5m) and maximum groundwater drawdown is reset to 10m to avoid much
more infeasible solutions in the population which probably inhibits the convergence of the
optimization. The hypervolume metric (HV) is used to evaluate the convergence of
Pareto-optimal solutions under the three scenarios. The advantage of HV is the monotonically
increasing relationship between the metric value and Pareto dominance, which shows the
optimal tradeoff surface can achieve maximum hypervolume (Bader and Zitzler, 2011). Fig. 12
shows all Pareto-optimal solutions in the four-dimensional objective space under different
runoff change scenarios. It is obviously observed that the tradeoff surface with current runoff
(Scenario A0) is closest to the ideal solution and those with runoff reduction are farther from
the solution. Scenario A2 based solutions exhibit worst performance owing to the greatest
extent of runoff reduction. Moreover, we rescaled the objective range to the interval [0, 1] and



set the reference point to the objective vector (1, 1, 1, 1) to calculate the HV metric. Fig. 13
shows the evolution of HV and the number of generation. Judged from the performance
evolution, tradeoff solutions under Scenario A0 achieve the largest HV and those in Scenario
A2 have the lowest HV that shows the solutions are far away from the Pareto-optimal front.
Therefore, the exploitation extent of surface diversion and groundwater abstraction should be
diminished in the face of runoff reduction derived from climate change. In Fig. 12,
Pareto-optimal solutions in Scenarios A1 and A2 does not exists when $f_{SRI}$ is greater than a
certain value and the diversity of solutions is obviously decreased. The reason is that
augmenting $f_{TWS}$ causes more decline of $f_{SRI}$ and the lake level compared with no reduction in
runoff in Scenario A0, which more likely generates a large amount of unfeasible solutions
violating the constraint of minimum lake level. The finding also shows that runoff in Kaidu
River through YB is a dominant factor controlling the variation of Bosten Lake level. To
investigate the effect of runoff reduction on the environmental benefits, Fig. 14 shows the
non-dominated fronts in the $f_{GSC}$ and $f_{SRI}$ objectives space across Scenarios A0, A1 and A2. The
solutions in Scenario A2 are completely dominated by the solutions in Scenarios A0 and A1.
Scenario A0 based solutions show the best Pareto optimality. Therefore, the runoff reduction
results in dramatic loss of environmental benefits. It is noteworthy that $f_{SRI}$ with Scenarios A1
and A2 will be reduced under the similar $f_{GSC}$. In the optimization, in order to maximize
irrigation water supply, sustaining similar groundwater storage in Scenarios A1 and A2 has to
be at the cost of river runoff decline to augment surface water diversion. Consequently, it is
essential for water managers to realize the conflict of conjunctive use of SW and GW for the
water management in arid inland basin.
**Figure 12.**
**Figure 13.**
**Figure 14.**



## 5. Conclusions

The study proposed a multi-objective optimization framework for the integrated surface water and groundwater management and demonstrated its effectiveness through a spatial optimization of water use practices for the agricultural irrigation in Yanqi Basin, a typical arid inland basin in northwest China. The well-calibrated simulation model with MODFLOW-NWT was developed to model the interaction of surface water (*i.e.*, Kaidu River and Bosten Lake) and groundwater. Then this study presented a new MOEA (the epsilon multi-objective memetic algorithm, $\varepsilon$-MOMA) and linked it with the numerical model to solve the multi-objective management model. The optimization model is composed of the four conflicting objectives: maximizing total water supply rate, minimizing total cost of transporting water from water intake points to water use destinations, maximizing the groundwater storage change in the aquifer and maximizing the surface runoff inflow from Kaidu River to Bosten Lake. An interactive visualization tool was applied to explore 4-dimensional tradeoff surface in a global view. Results showed augmenting water supply caused the larger cost of water delivery, reduced the runoff inflow to lake and aggravated the loss of groundwater storage. The 2-dimensional compromise schemes selected from the non-dominated fronts between $f_{TWS}$ and other objectives exhibited significant decision bias in the higher order objective spaces. Therefore, it is essential for decision-makers to explore water management schemes in the many-objective tradeoff surface.

The 4-dimensional compromise solutions were obtained to investigate performance of existing scheme. Result showed the water use practices before optimization must be regulated to avoid unnecessary capital expenditure of transporting water. However, the compromised solution indicated groundwater storage was still decreasing. Thus, the water managers may be inclined to adopt the Pareto-optimal scheme satisfying minimum water demands to prevent the loss of groundwater storage and runoff inflow to the lake. In the practical application, the





decision-makers should identify specific irrigation water demands and environmental
constraints to discover preferred water use schemes. The scenarios of runoff change were
created to investigate the effect of runoff reduction in Kaidu River on the regional water
resources management. The findings showed that reducing runoff inflow to YB could lead to
the degradation of Pareto solutions compared with those based on the current runoff scenario.
In this light, it is of crucial importance to implement stringent water management schemes and
explore potential water-saving strategies in the face of the uncertainty.
The findings in the study are essential to regional water resources management in a typical
arid inland basin with long-term intensive agricultural development. However, due to the
data-scarcity in the basin-scale water cycle and limitations of simulation model, the current
model may be not enough to reflect the complex relationship in the groundwater-river-lake
hydrological system. Future research should focus on exploiting fully coupled numerical model
to accurately simulate basin-scale water cycle and avoid decision bias derived from the
numerical model. Meanwhile, deep uncertainty (*e.g.*, land use change, climate change, etc.) is a
key factor to affect the robustness and reliability of the optimal solutions in the changing world.
In the simulation-optimization framework, integrating these factors into the management
model to explore optimal schemes is a research focus in the future.
**Acknowledgements**
This study is jointly supported by the National Natural Science Foundation of China
(41730856 and 41772254) and the National Key Research and Development Plan of China
(2016YFC0402800). The numerical calculations in this study have been implemented on the
IBM Blade cluster system in the High Performance Computing Center of Nanjing University.

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





**Tables**
Table 1 The control parameters of $\varepsilon$-MOMA and epsilon value of objectives

| Parameter | Value |
|---|---|
| Population size ($N_{pop}$) | 200 |
| Maximum function evaluation ($N_{eval}$) | $6\times10^4$ |
| Crossover probability ($P_c$) | 0.90 |
| Mutation probability ($P_m$) | 0.05 |
| $f_{TWS}$ epsilon (m$^3$/yr) | $1\times10^4$ |
| $f_{TCOST}$ epsilon (CNY/yr) | $1\times10^2$ |
| $f_{GSC}$ epsilon (m$^3$/yr) | $1\times10^4$ |
| $f_{SRI}$ epsilon (m$^3$/yr) | $1\times10^4$ |







Table 2 The objective values corresponding to several solutions

| Objective | Solution 4 | Solution 5 | Solution 6 | Solution 7 |
|---|---|---|---|---|
| $f_{TWS}$ (×$10^8$ m$^3$/yr) | 10.7406 | 12.7355 | 8.6712 | 10.1032 |
| $f_{TCOST}$ (×$10^6$ CNY/yr) | 54.3013 | 92.1498 | 42.9522 | 82.7827 |
| $f_{GSC}$ (×$10^8$ m$^3$/yr) | -0.2471 | -1.2856 | 0.2192 | -0.4462 |
| $f_{SRI}$ (×$10^8$ m$^3$/yr) | 17.5698 | 17.0030 | 17.8180 | 17.1880 |







**Figures**

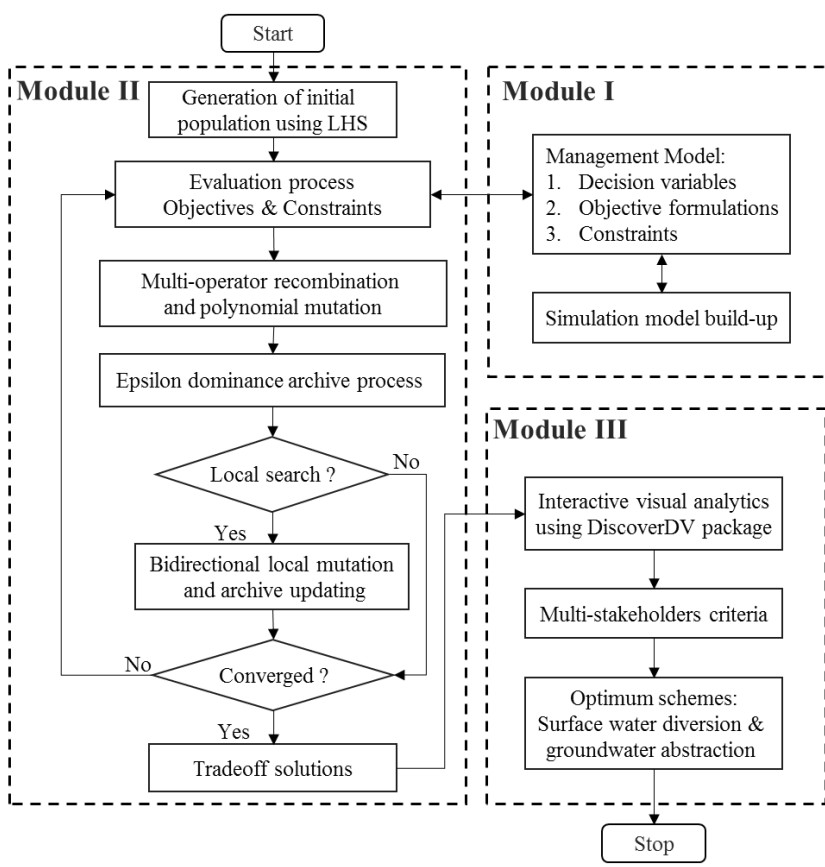


**Fig. 1.** Framework of multi-objective optimization for integrated SW-GW management.



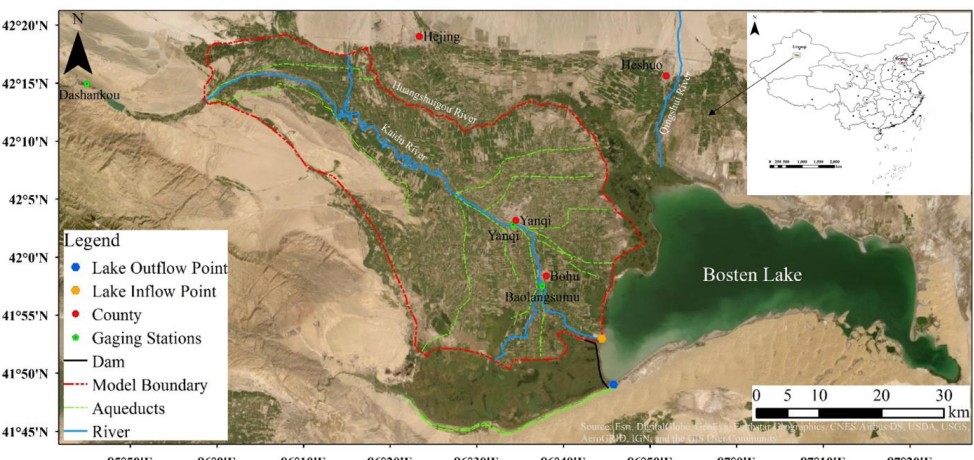

**Fig. 2.** The location of Yanqi Basin and the model domain of interest for this study. Source: DigitalGlobal, Inc. (imagery).





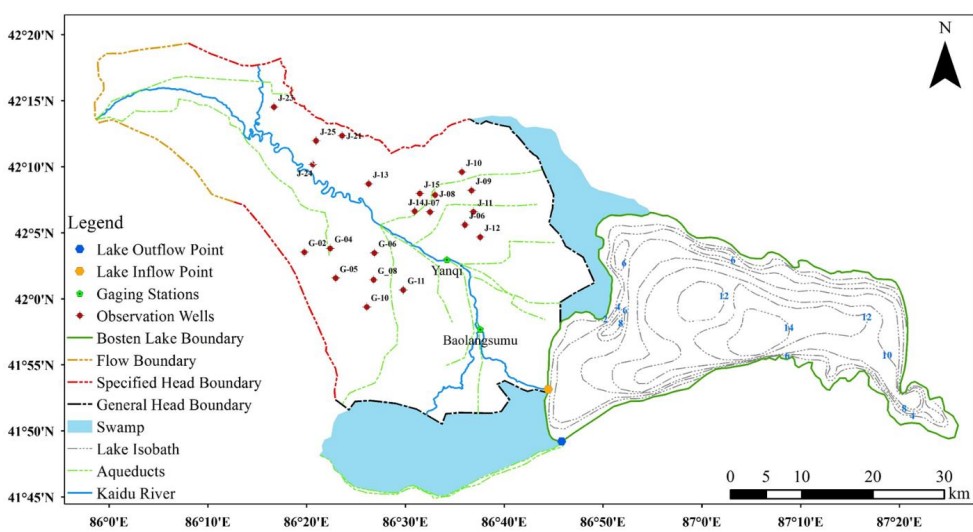


**Fig. 3.** The boundary conditions of model domain, monitoring locations of groundwater level

and surface runoff, aqueduct system and bathymetric contours in meters for Bosten

Lake.



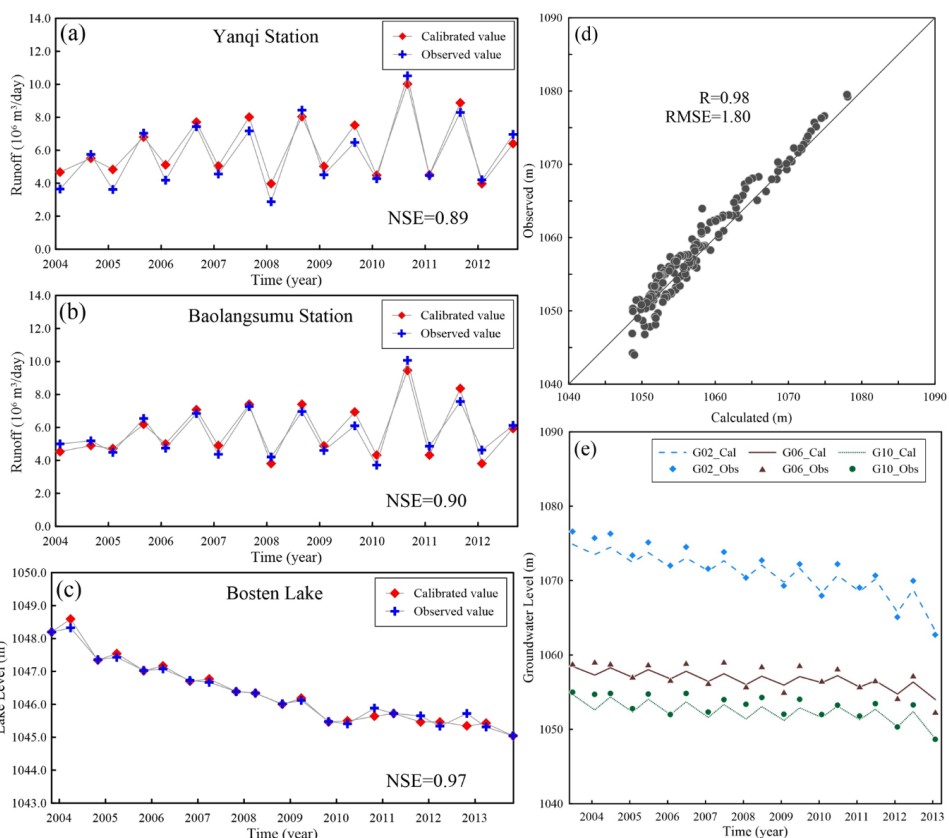

**Fig. 4.** The calibrated results of the transient model showing (a) observed vs. calibrated runoff at Yanqi station over time, (b) observed vs. calibrated runoff at Baolangsumu station over time; (c) observed vs. calibrated lake level over time; (d) comparison of observed and calibrated groundwater heads at all observation wells, and (e) observed vs. calibrated groundwater heads over time at three typical observation locations as labeled in Fig. 3.





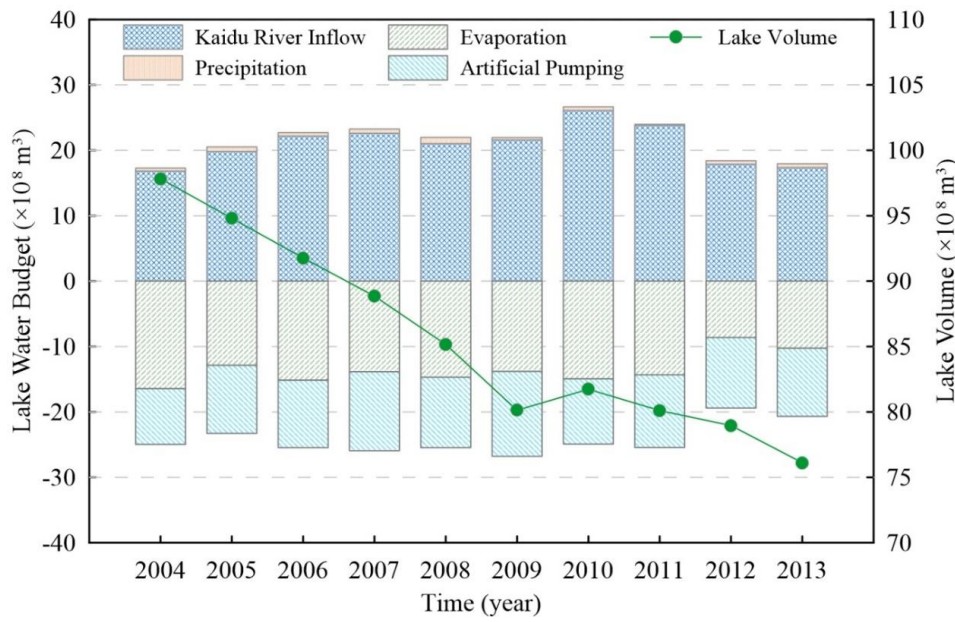

**Fig. 5.** The water balance terms of Bosten Lake and resulting lake volume in the simulation period.





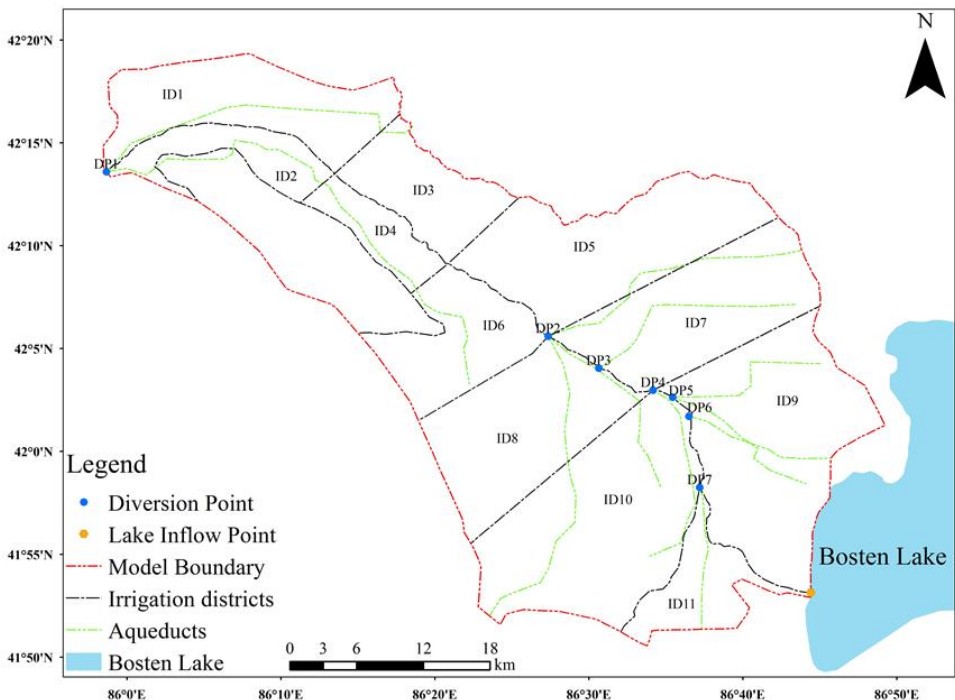


**Fig. 6.** The locations of surface water diversion points and subdomains of irrigation districts for groundwater abstraction.







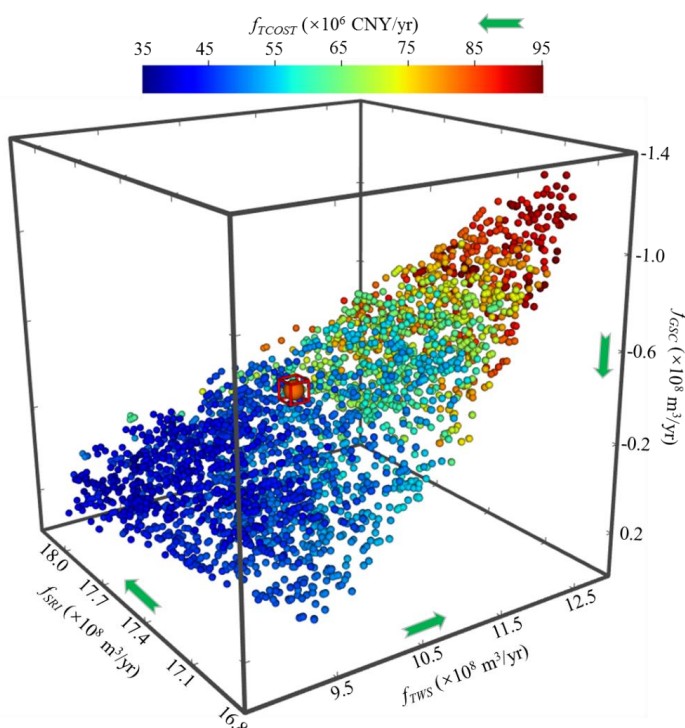


**Fig. 7.** The tradeoff surface to the integrated SW-GW management in Yanqi Basin. Each spheric symbol represents a water use scheme corresponding to specific objective values of the total water supply rate ($f_{TWS}$), total cost of water delivery ($f_{TCOST}$), surface runoff inflow to lake ($f_{SRI}$) and groundwater storage change ($f_{GSC}$). $f_{TCOST}$ is symbolized in color to identify the objective value against others. The green arrow is the direction of better performance for each objective. The scheme before optimization is marked in a red square box.






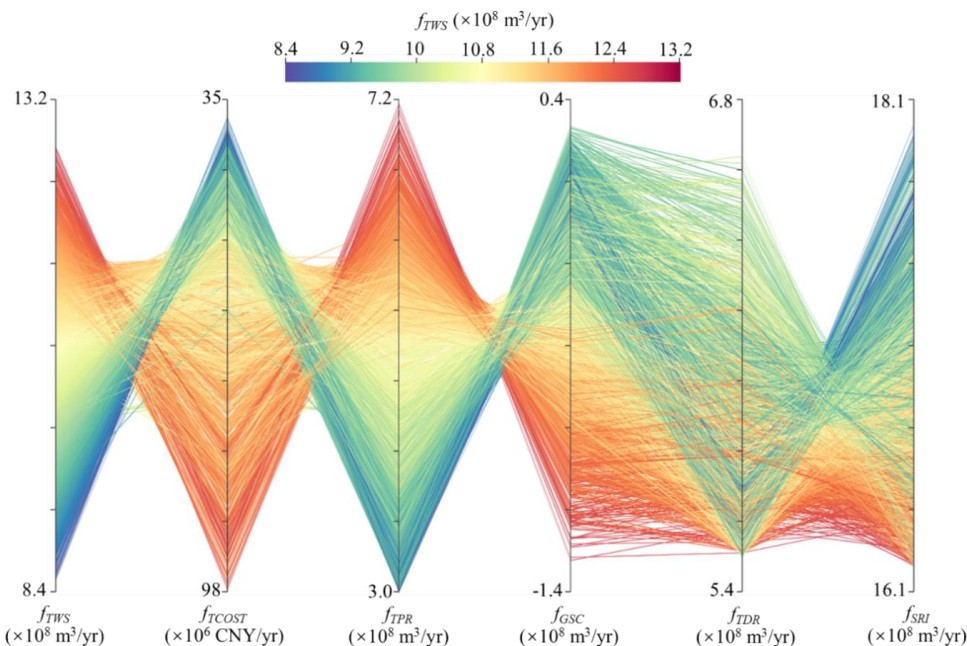


**Fig. 8.** The objective values (*y*-axis) are plotted over management objectives $f_{TWS}$, $f_{TCOST}$, $f_{GSC}$, $f_{SRI}$, total pumping rate $f_{TPR}$ and total surface water diversion rate $f_{TDR}$ (*x*-axis), $f_{TWS}$ is represented in color. The preferred direction for each index is upward.







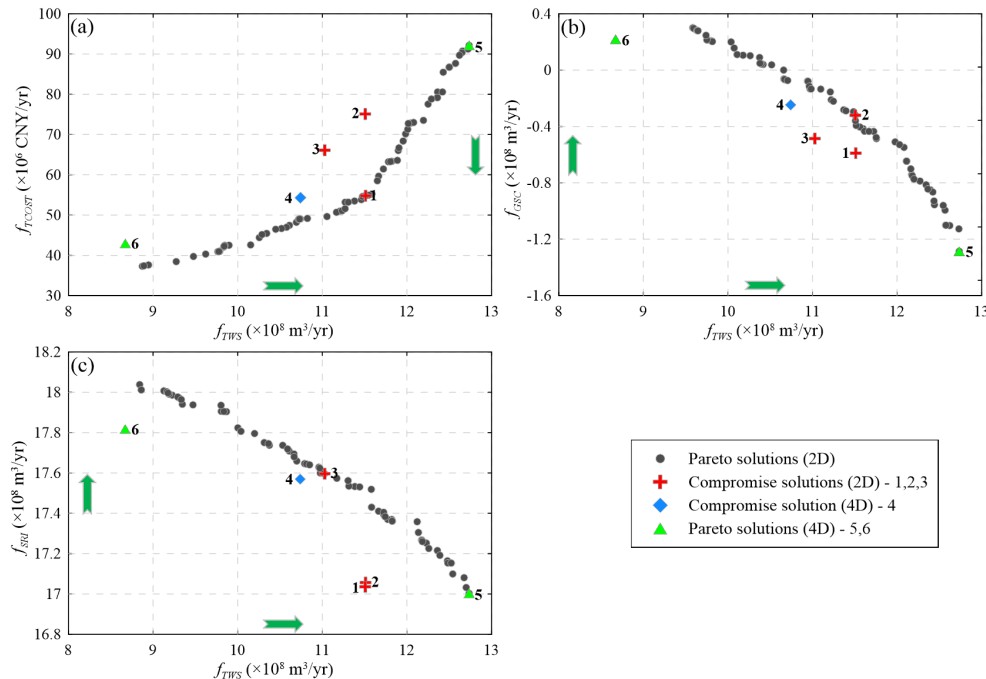

**Fig. 9.** Identification of six interesting solutions (Solutions 1-6) from the four-dimensional approximate Pareto set and the green arrow is the preferred direction for each objective.



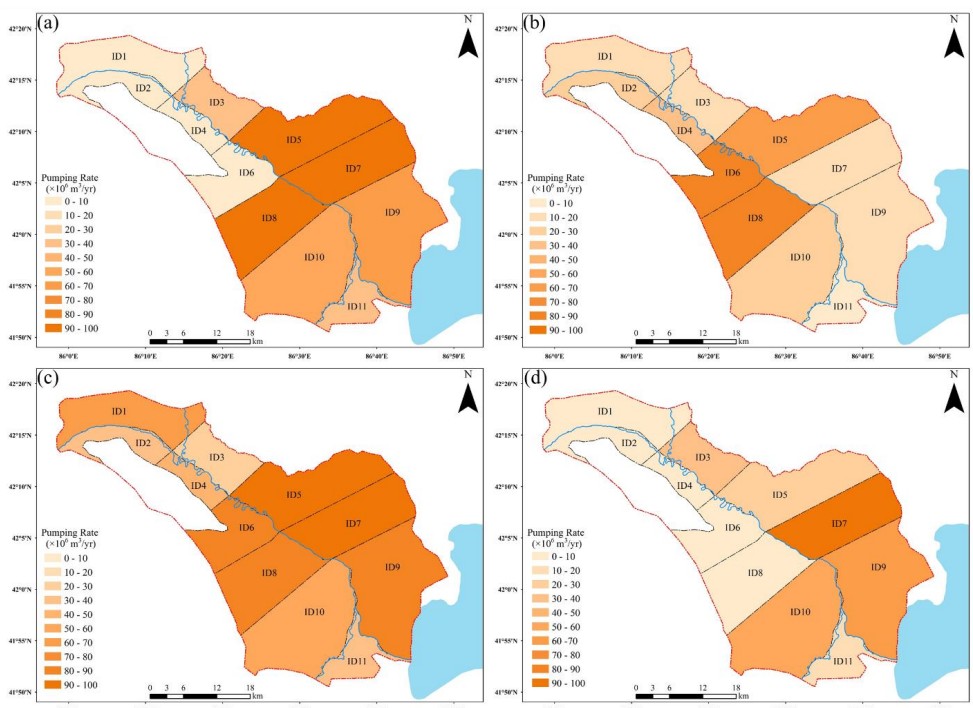


**Fig. 10.** The spatial distribution of the pumping rates in the 11 irrigation districts for the four
selected schemes of (a) Solution 4, (b) Solution 7, (c) Solution 5, and (d) Solution 6,
respectively.





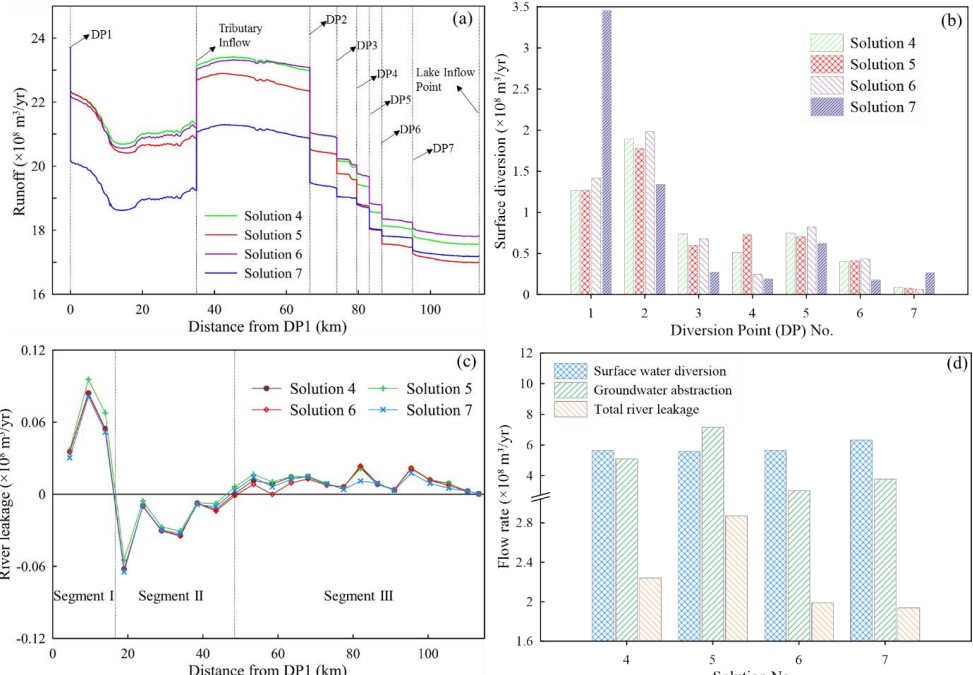


**Fig. 11.** Variation of surface runoff and river leakage along the stem stream of Kaidu River: (a)
the profile of river runoff; (b) the distribution of surface water diversion at the different
diversion points; (c) the profile of river leakage; (d) the components of total river
leakage, groundwater abstraction and surface water diversion for several typical
Solutions 4-7.






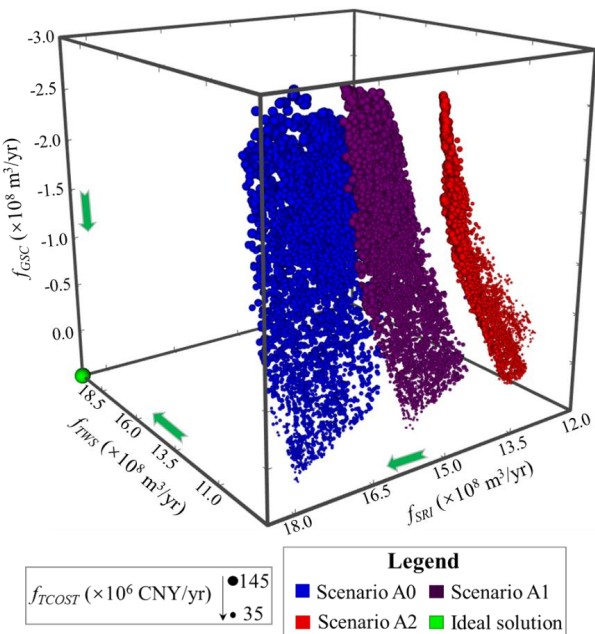


**Fig. 12.** The tradeoff solutions under Scenarios A0, A1 and A2, and the sphere size indicates
the value of $f_{TCOST}$. The green arrow is the direction of better performance for each
objective.







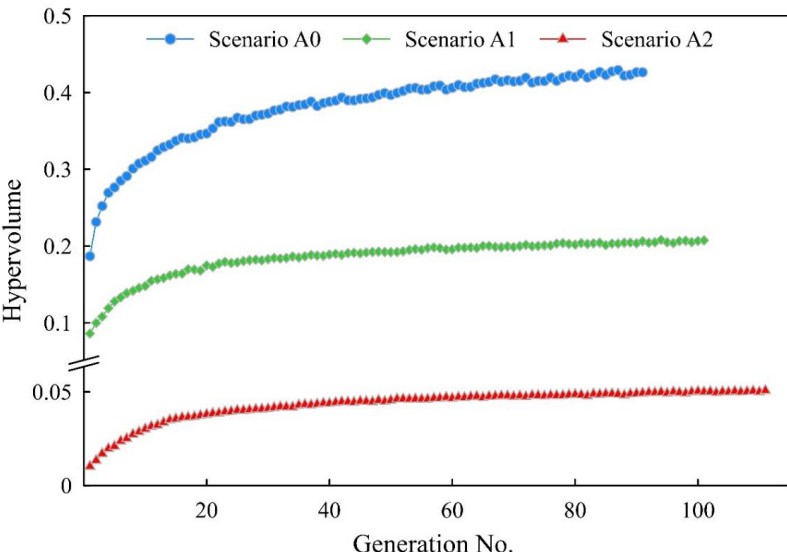

**Fig. 13.** Evolution of the hypervolume metric over the generation number for Scenarios A0, A1 and A2.





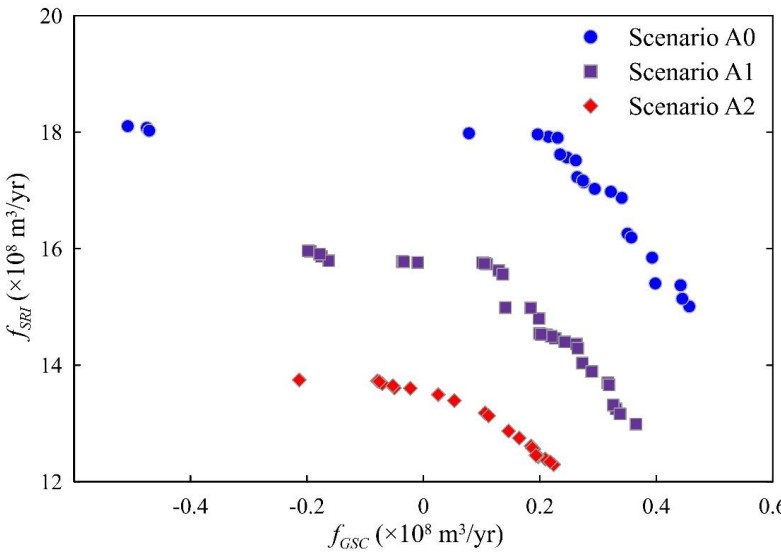

870

**Fig. 14.** Non-dominated fronts of Scenarios A0, A1 and A2 between objectives of $f_{GSC}$ vs. $f_{SRI}$.

872

873