# Peer review of "Basin-scale multi-objective simulation-optimization modeling for"

_Hydrology and Earth System Sciences, 2019_

## Referee Comment (RC1) · Joseph Kasprzyk (Referee) · 11 Jul 2019

I am serving as a requested referee for this manuscript. The paper presents a new optimization algorithm, linked to hydrological models for the purpose of informing water management in China. Overall, the paper does provide an interesting case study. However, the authors could do a better job of contextualizing their work relative to the state-of-the-art literature in this field, summarized in my general comments below. I also provide specific comments referencing lines in the manuscript itself.

[Figure]

General comments:

I. The need for a new MOEA should be justified. Moreover, since MOEAs are typically designed for general case studies outside of water management, the authors should indicate whether the new algorithm is available for use.

II. I would like to see more description of the optimization in general, since the calibration of hydrological models is not really the focus of the analysis.

III. The results should be generalizable to a broader context. What are the take-home messages for the HESS audience? This is hinted at in the Conclusion, but could be better motivated in the Introduction.

Specific comments, where line numbers refer to the PDF version of the HESSD paper:

1. The authors should consider editing lines 25-29 to clarify the novelty of the paper. A study of one basin in China may not be compelling to an international audience, so if there is something new about the coupling of optimization to model, that should be highlighted. The same comment is relevant for the introduction; the scientific contribution of the paper is not sufficiently stated.

2. line 48: There are several grammatical errors in the beginning of the paper ("In arid and semi-arid basin,") as well as a disconnect between talking about water management in general and moving quickly to the specifics of China. A native English speaker should proofread the manuscript throughout.

3. line 73-74: "to tackle intricate SW and GW management model": Is this a typo? I think the intended word might be "SW and GW management problems". Also "tackle" is probably not an appropriate word to use.

4. line 77: Before the first mention of "bi-objectives", the authors should provide a very brief introduction to optimization. Otherwise, readers may be confused by what is meant by "objective" throughout this paragraph.

5. When introducing MOEAs, it would be good to cite Maier et al (2019), which is an introductory overview appropriate for readers to be introduced to the topic.

6. lines 102-113: I am glad the authors have brought up some recent and relevant topics in many-objective optimization. However, the paragraph was confusing and will be difficult for readers to follow. For example, the Borg algorithm is briefly mentioned, but there is no clear transition to the next algorithm ("In order to enhance the local optimality..."). Did Sindya et al. add to Borg or create a new algorithm? Moreover, it is unclear whether the authors made a new algorithm, and whether it builds on the work of Hadka and Reed and Sindhya. Moreover, given that these new algorithms have been extensively tested (e.g., Reed et al 2013), it is worth justifying why a new algorithm is needed.

7. line 114: The section would benefit from a better transition between the MOEA material and the GW modeling material. Also, since the GW modeling is being done in the context of decision making, I would like to see a clearer discussion of the decision variables and objectives of the optimization as the problem is being introduced.

8. line 125: Ideally, a paragraph would express one idea at a time. Here, the authors have transitioned from discussing their method to providing details about their case study. This material should be separated.

9. line 168-169: What is meant by "decision makers" here? In many systems, different people make decisions about the irrigation diversions, lake storage, and groundwater pumping. Without a clear context for decision making, this section is too vague.

10. line 179-186: There is some repetition here compared to the introduction. Although I agree with the points about nonlinearity, nonconvexity, etc., it is more useful at this point in the paper to explain the details of the proposed new algorithm.

11. Is this the first introduction of the e-MOMA algorithm? If not, it would be very helpful to have a citation to the original reference, since there is not enough detail given here.

[Figure]

At the least, the authors should justify how their algorithm differs from Hadka and Reed, and others.

12. line 253: The discussion of the Ecological Water Conveyance Project is interesting. I'd like to see it integrated better within the text. Is this study supporting that analysis?

13. Equations 3-5: Why were different metrics used for different variables?

14. line 322-323: This statement should be justified. It speaks to the wider question of how the hydrological modeling is serving the ultimate goal of the management problem, as well as the general contribution of the paper itself. If the focus of the paper is too diffused, it becomes hard to follow its details.

15. line 348: To what extent can the groundwater extraction rate be controlled? In some systems, farmers have juristiction on how much to pump. If there is an implicit assumption about a set of water managers who can dictate water usage, this should be stated.

16. lines, 353, 357, etc.: Is there a citation to the water price data? Or was this just an assumption?

17. line 406: Guidance on interpreting parallel plots should be provided.

18. line 534: When the authors say "This study implemented...", were they implying that this occurred across the entire study? Or only in one part of the study? This should be clarified.

19. line 541-542: My impression is that hypervolume analyses are usually done to compare optimization runs with the true Pareto set. Is this known? In general, since the optimization seems to be the focus of this paper, items such as Hypervolume Analysis should be covered in the Methodology (which means that some hydrological modeling detail can be removed)

20. line 546: "obvious" is usually not appropriate in technical writing.

21. Conclusion section: The quality of writing here is much better than in the introduction. Some of this material should inform the Introduction, since this more clearly articulates the purpose of the study than the beginning of the paper did.

22. In spite of comment #21, I would like to see slightly more discussion about the management implications of this study – in the local case study as well as how the results can be transferred to other basins (especially given different legal and regulatory structures).

23. Table 1: Was random seed analysis performed? If so, the parameters of this analysis should also be provided here. The epsilon values seem quite small – were larger epsilons attempted?

24. Figure 4: If the paper is too long, I could imagine this figure could go into supplemental material. Also, I noticed that the NSE values appear in the figure but were not referenced in the text.

25. Figure 9: If possible, the other solutions that are Pareto optimal in 4 dimensions but not in two, should be shown on this plot. Otherwise, the idea that the highlighted solutions fall "outside the front" will be confusing to readers. The Kollat and Reed (2007) paper referenced in this manuscript shows how to do this.

26. Figure 12: The figure would be easier to understand if the authors reminded reader what these scenarios represent (see also comment #18 – the scenario analysis could be better explained overall).

References

Maier, H.R., Razavi, S., Kapelan, Z., Matott, L.S., Kasprzyk, J., Tolson, B.A., 2019. Introductory overview: Optimization using evolutionary algorithms and other metaheuristics. Environmental Modelling & Software 114, 195–213. https://doi.org/10.1016/j.envsoft.2018.11.018

Reed, P.M., Hadka, D., Herman, J.D., Kasprzyk, J.R., Kollat, J.B., 2013. Evolutionary Multiobjective Optimization in Water Resources: The Past, Present and Future. Advances in Water Resources 51, 438–456.

---

## Author Comment (AC1) · 4 Aug 2019

In the supplement (ZIP file), we have fully addressed Referee Dr. Joseph Kasprzyk's concerns in the marked PDF file and given a point-by-point response in the PDF file (Response to Referee.pdf).

Please also note the supplement to this comment:
https://www.hydrol-earth-syst-sci-discuss.net/hess-2019-278/hess-2019-278-AC1-supplement.zip

---

## Short Comment (SC1) · 14 Sep 2019

This manuscript focus on the topic of conjunctive using of surface water and groundwater based on the multi-objective simulation and optimization method. In the manuscript, a novel multi-objective optimization model with four objective functions is developed to balance the water demand for agriculture, socioeconomic development and environmental demands. In order to find the Pareto optimal solutions for the special model, a new multi-objective evolutionary algorithm, named ÉŻ-MOMA, is presented. The optimization results of Yanqi Basin (YB) in northwest China certified the applicability of

the new model and optimization algorithm. Generally, the manuscript does make an important contribution on water resources management research. However, there are some general and specific comments referencing lines in the manuscript which will be helpful for the improvement of the manuscript.

General comments: 1. The introduction usually includes the research background, the research problems, a review of the advantages and disadvantages of the previous and latest research results, and the new solving method of the present research. Thus, the description of the detailed condition of the study area should be move into section 3.1. 2. The advantage of the newly developed optimization algorithm should be given in detail. For example, why the ÉŻ-MOMA is better than the other MOEA in solving groundwater management problems? What is the main difference between ÉŻ-box and the elite individual preservation strategy? 3. In the numerical simulation process, the irrigation backflow should be considered. How to deal with the irrigation backflow in the groundwater flow numerical simulation model of the YB? 4. YB is a typical arid inland basin in China. The optimization results of YB are seemed reliably, can this optimization model be used directly in other basin or field?

Specific comments: 1. Line 168: I suggest to changing the "decision-maker" to "water manager" in the manuscript. The author sometimes uses "decision-maker" and sometimes uses "water manager", which will confuse the readers. 2. Line 199-201: Did all of the referred recombination operators (SBX, DE, SPX, PCX, LX, UM) used in the new optimization method? Or only one of them was adopted? The author should clear it. 3. Line 291: There is a mistake in Equation 4, the "2" was lost. 4. Line 417: where the increment of fTPR and fTDR from, the explanation should be given. 5. Line 539-541: Why the lake level is changed to a smaller value? And why the maximum groundwater drawdown is reset to 10m? 6. Line 557: "a certain value" should be given explicitly for the case study. 7. Line 578: Change "Yanqi Basin" to "YB".

[Figure]

278, 2019.

---

## Referee Comment (RC2) · Anonymous Referee #2 · 23 Dec 2019

This paper developed a multi-objective simulation-optimization framework for sustainably conjunctive use of surface water and groundwater and applied it to water allocation in Yanqi Basin, an arid region in northwest China. The framework employed the epsilon multi-objective memetic algorithm with the MODFLOW-NWT based simulation model and used four management objectives in their optimization. The final results are very useful for sustainable water management in the study area and provided useful support to decision makers for water allocation. This paper can be suggested for possible publication in HESS after taking carefully into account the comments listed below.

[Figure]

Specific comments:

1) This study developed a multi-objective simulation-optimization framework for sustainably conjunctive use of surface water and groundwater. I didn't really see the new insights the readers can get, if only from the introduction part of this manuscript. Can the authors clarify the differences between their work and others? Partially solving the domination resistance phenomenon seems not new. Adding an epsilon to MOMA seems not new as well. 2) You mentioned SFR2, LAK3 and MODLOW-NWT. Since this is an important part of the framework, can you add more details about these simulation models in the revised manuscript? For example, SFR2 is a streamflow-routing package. Does this model include hydrological simulation, or just a hydraulic model since it is only named as a routing model?How was MODLOW-NWT developed in the study area? 3) Was the epsilon MOMA algorithm developed by yourself? The references attached are not enough to understand the algorithm. Please add more details about the logic line of the algorithm. 4) Figure 2: the figure didn't show clearly the river names in the basin. For example, I cannot find Kongqi River and lower Tarim River in the figure. This figure should help us understand the rivers, aqueducts etc. Please add a more detailed map. 5) When setting up the simulation model, what kind of data and also the details of data should be explained. What data were used for model calibration and validation? 6) What is "stress period"? 7) What is the time resolution in your simulation model? From Fig.4, you can see that the resolution is very coarse, semi-annually? This model fails to show even the seasonality of runoff, lake level and water allocation. 8) How did the simulation consider all human activities in the model? For example, how SFR2 take into account the diversion or abstract of water from the river? 9) Page 17: How did you obtain the scheme before optimization? 10) Climate change has substantial impacts on river runoffs in arid rivers in Xinjiang Province. The authors used only three simple scenarios (Current runoff; reduce 10% runoff and reduce 20% runoff) to investigate the impacts of climate change. These scenarios are just toys and don't provide useful information for climate change adaptation for the study area. Why didn't the authors use more practical climate change scenarios like RCPs? 11) Pos-

sible uncertainty in the simulation-optimization model and decision making should be discussed in the manuscript.

Technical corrections: Only minor typo is found.

---

## Author Comment (AC3) · 10 Jan 2020

The comment was uploaded in the form of a supplement:
https://www.hydrol-earth-syst-sci-discuss.net/hess-2019-278/hess-2019-278-AC3-supplement.zip

---

## Author Response (AR1)

Response to the comments on the manuscript (**HESS-2019-278**)
"**Basin-scale multi-objective simulation-optimization modeling for conjunctive use of surface water and groundwater in northwest China**"
by Jian Song, Yun Yang, Xiaomin Sun, Jin Lin, Ming Wu, Jianfeng Wu, and Jichun Wu

Note that the following text in color Times New Roman denotes Referee's comments and in Times New Roman font denotes our response to the comments in the discussion. In our resubmission, the marked PDF file combined with the response file has clearly indicated all changes to the original manuscript, tables and figures. Also, in the marked PDF file, marked in a blue strikethrough font is the text that should be removed from the original manuscript and marked in a red font is the text that has been added to the revision. In addition, Line number(s) mentioned below can be referred to as that line numbering in the marked revised manuscript.

**Response to Referee #1 Dr. Joseph Kasprzyk's Comments**

I am serving as a requested referee for this manuscript. The paper presents a new optimization algorithm, linked to hydrological models for the purpose of informing water management in China. Overall, the paper does provide an interesting case study. However, the authors could do a better job of contextualizing their work relative to the state-of-the-art literature in this field, summarized in my general comments below. I also provide specific comments referencing lines in the manuscript itself.

**[Response]** We appreciate Dr. Joseph Kasprzyk's insightful comments and constructive suggestions. We have fully addressed his concerns into the revised manuscript and given a point-by-point response as below.

General comments:

I. The need for a new MOEA should be justified. Moreover, since MOEAs are typically designed for general case studies outside of water management, the authors should indicate whether the new algorithm is available for use.

**[Response]** Indeed, the several state-of-the-art MOEAs (*e.g.*, $\varepsilon$-NSGAII (Kollat et al., 2006), MOEA/D (Zhang and Li, 2007), NSGAIII (Deb and Jain, 2014), Borg (Hadka and Reed, 2013)) have been tested on the standard test problems even on the real-world problems and achieved the promising results in solving many-objective problems. However, due to the diversity and complexity of real-word decision-making problems, efforts should be made to develop the advanced MOEAs (**Lines 126-133**). Moreover, the aim of our research is to construct an effective many-objective optimization framework for water resources management in arid inland basin rather than implement comparative study of the state-of-the art MOEAs to justify the optimality of the algorithm.

On the other hand, we acknowledge that the performance of $\varepsilon$-MOMA has to be tested to prove the availability for general case studies. Considering the reviewer's concerns, we have investigated the performance of $\varepsilon$-MOMA by the benchmark test problems in **Section 2.2.2** (*i.e.*, 3 to 6-objective DTLZ1 and DTLZ3 problems) (Deb et al., 2002). The results show that $\varepsilon$-MOMA can provide reliable and diverse Pareto-optimal solutions to many-objective optimization problems (**Table S1** in the Supplementary Materials). Meanwhile, the basin-scale case study of this paper further shows the potential of $\varepsilon$-MOMA for the real-world water resources management.

Deb, K., Jain, H.: An evolutionary many-objective optimization algorithm using reference-point-based nondominated sorting approach, Part I: solving problems with box constraints, IEEE Trans., 18(4), 577-601, https://doi.org/10.1109/TEVC.2013.2281535, 2014.

Deb, K., Thiele, L., Laumanns, M., Zitzler, E.: Scalable multi-objective optimization test problems, in: proceeding of the congress on evolutionary computation (CEC-2002), 825-830, 2002.

Hadka, D., and Reed, P.M.: Borg: an auto-adaptive many-objective framework, Evol. Comput, 21(2), 213-259, https://doi.org/10.1162/EVCO_a_00075, 2013.

Kollat, J. B., Reed, P. M.: Comparing state-of-the-art evolutionary multi-objective algorithms for long-term groundwater monitoring design, Adv. Water Resour., 29(6), 792-807, https://doi.org/10.1016/j.advwatres.2005.07.010, 2006.

Zhang, Q., Li, H.: MOEA/D: A multiobjective evolutionary algorithm based on decomposition, IEEE Trans. Evol. Comput., 11(6), 712-731, https://doi.org/10.1109/TEVC.2007.892759, 2007.

II. I would like to see more description of the optimization in general, since the calibration of hydrological models is not really the focus of the analysis.

[**Response**] Comment accepted. We have made detailed explanations to present the algorithmic process step by step in **Section 2.2.1**. The hydrological model, as a prerequisite for the simulation-optimization method, has to be calibrated to reflect the responses of water resources system under the management schemes. Considering the referee's concern, we have briefly stated the calibrated results of key state variables in **Section 3.2** and put the results in the Supplementary Materials as shown in **Fig. S2**. Moreover, the analysis of water balance in Bosten Lake paves the way for the construction of management model.

III. The results should be generalizable to a broader context. What are the take-home messages for the HESS audience? This is hinted at in the Conclusion, but could be better motivated in the Introduction.

**[Response]** The point is well taken. The study results show that Pareto-optimal solutions considering environmental and socioeconomic factors can be achieved for the basin-scale water resources management involving complicated groundwater-river-lake interactions. Meanwhile, due to the water scarcity and climate change, the conservative water management options may be implemented to sustain the fragile ecosystem in the arid inland basin. Considering reviewer's concerns, we have added necessary explanations in the section "Introduction" to present the motivation (**Lines 53-65**) and the general results (**Lines 196-202**).

Specific comments, where line numbers refer to the PDF version of the HESSD paper:

1. The authors should consider editing lines 25-29 to clarify the novelty of the paper. A study of one basin in China may not be compelling to an international audience, so if there is something new about the coupling of optimization to model, that should be highlighted. The same comment is relevant for the introduction; the scientific contribution of the paper is not sufficiently stated.

**[Response]** Comment accepted. We appreciate the reviewer's insight and have modified the statement in Abstract (**Lines 24-33**) and the section "Introduction" (**Lines 176-181**) in the revised manuscript to clearly present the contribution of the study.

2. Line 48: There are several grammatical errors in the beginning of the paper ("In arid and semi-arid basin,") as well as a disconnect between talking about water management in general and moving quickly to the specifics of China. A native English speaker should proofread the manuscript throughout.

**[Response]** Comment accepted. We have modified the statements (**Line 53**). To avoid the disconnect raised by the referee, we have reorganized the statements in the Introduction. First, we have clarified the need for water management in the arid inland basin (**Lines 53-63**). Second, we explained the meaning of many-objective optimization framework in the water resources management and planning (**Lines 81-109**) and the optimization techniques (**Lines 110-137**). After that, we introduced the specifics of water resources development in Yanqi Basin (**Lines 144-160**) to explain the suitability of the case study. As for the language problem, a native English speaker is difficult for us to find. However, in the revised manuscript, one of the co-authors who ever worked as a visiting scholar in the USA for several years has made extra efforts in current revision to correct the grammatical and wording errors.

3. Line 73-74: "to tackle intricate SW and GW management model": Is this a typo? I think the intended word might be "SW and GW management problems". Also "tackle" is probably not an appropriate word to use.

**[Response]** Comment accepted. We have modified the statement as "in solving the complex SW and GW management problems" in the revision **(Lines 90-92)**.

4. Line 77: Before the first mention of "bi-objectives", the authors should provide a very brief introduction to optimization. Otherwise, readers may be confused by what is meant by "objective" throughout this paragraph.

**[Response]** The point is well taken. We have added necessary explanations to briefly introduce the process of simulation-optimization approach **(Lines 82-86)**.

5. When introducing MOEAs, it would be good to cite Maier et al (2019), which is an introductory overview appropriate for readers to be introduced to the topic.

**[Response]** Comment accepted and change made as suggested **(Lines 110-113)**.

6. Lines 102-113: I am glad the authors have brought up some recent and relevant topics in many-objective optimization. However, the paragraph was confusing and will be difficult for readers to follow. For example, the Borg algorithm is briefly mentioned, but there is no clear transition to the next algorithm ("In order to enhance the local optimality..."). Did Sindya et al. add to Borg or create a new algorithm? Moreover, it is unclear whether the authors made a new algorithm, and whether it builds on the work of Hadka and Reed and Sindhya. Moreover, given that these new algorithms have been extensively tested (e.g., Reed et al 2013), it is worth justifying why a new algorithm is needed.

**[Response]** In the revised manuscript, we have firstly stated the difficulty in the many-objective optimization (*i.e.*, the domination-resistance phenomenon) **(Lines 113-116)**. Then we presented two kinds of state-of-art MOEAs by which an attempt to alleviate the difficulty is feasible **(Lines 116-126)**. Finally, we proposed a new MOEA, named epsilon multi-objective memetic algorithm (ε-MOMA), which utilized several advanced techniques from Borg MOEA and a local search operator to enhance the capacity of evolutionary search.

Sindya et al. (2013) proposed a hybrid framework for evolutionary multi-objective optimization and overcame some shortcomings of MOEAs (*e.g.*, slow convergence, inefficient termination criterion). In this study, we cited the work of Sindya et al. (2013) to show the efficiency of the hybrid framework (*i.e.*, memetic algorithm) for multi-objective optimization **(Lines 122-126)**.

As stated in **Lines 127-133**, the state-of-the-art MOEAs have been extensively used in the optimization problems, however, the complex real-world problems still show the deficiency of some advanced MOEAs. For example, Zheng et al. (2016) implemented the comparison of three MOEAs (NSGAII, SAMODE, Borg) in the water distribution system design. Results show NSGAII exhibits a more robust performance than other MOEAs. Borg converges quickly to the Pareto-optimal front whereas decreases the diversity of Pareto solutions.

As stated in the response to Dr. Kasprzyk's **General Comment I** above, this study has supplementarily exploited classical DTLZ problems to test the performance of $\varepsilon$-MOMA in **Section 2.2.2**. The optimization results show the potential of our algorithm (**Table S1** in the Supplementary Materials) and it would be further applied to solve basin-scale water resources management. Certainty, the performance of the algorithm needs to be validated on the challenging real-world problems, that's our focus in the future.

Sindhya, K., Miettinen, K., Deb, K.: A hybrid framework for evolutionary multi-objective optimization, IEEE Trans., 17(4), 485-511, https://doi.org/10.1109/TEVC.2012.2204403, 2013.

Zheng, F., Zecchin, A.C., Maier, H.R., Simpson, A.R.: Comparison of the searching behavior of NSGA-II, SAMODE, and Borg MOEAs applied to water distribution system design problems, J. Water Resour. Plann. Manage., 142(7), 04016017, https://doi.org/10.1061/(ASCE)WR.1943-5452.0000650, 2016.

7. Line 114: The section would benefit from a better transition between the MOEA material and the GW modeling material. Also, since the GW modeling is being done in the context of decision making, I would like to see a clearer discussion of the decision variables and objectives of the optimization as the problem is being introduced.

**[Response]** Comment accepted. We have made revisions in the paragraph to highlight the details of optimization model (**Lines 170-176**) and deleted the redundant statements of simulation model **(Lines 160-170)**.

8. Line 125: Ideally, a paragraph would express one idea at a time. Here, the authors have transitioned from discussing their method to providing details about their case study. This material should be separated.

**[Response]** Comment accepted and change made as suggested **(Lines 181-182)**.

9. Line 168-169: What is meant by "decision makers" here? In many systems, different people make decisions about the irrigation diversions, lake storage, and groundwater pumping. Without a clear context for decision making, this section is too vague.

**[Response]** In **Section 2.1**, our purpose is to state the general problem formulation for conjunctive management of surface water and groundwater in the arid inland basin. The decision makers in this study refer to the local water resources authority in the local government. Considering Referee Dr. Kasprzyk's concerns, we have made necessary revisions for the context of decision making in the revised manuscript (**Lines 228-231**).

10. Line 179-186: There is some repetition here compared to the introduction. Although I agree with the points about nonlinearity, nonconvexity, etc., it is more useful at this point in the paper to explain the details of the proposed new algorithm.

**[Response]** Comment accepted. To clearly state the algorithmic process and investigate the performance of the algorithm, we split the section into **Section 2.2.1** "Main algorithmic structure" and **Section 2.2.2** "Benchmark test". In **Section 2.2.1**, we have deleted the repetition (**Lines 243-248**) and presented process of the proposed algorithm step by step (**Lines 250-288**).

11. Is this the first introduction of the e-MOMA ($\varepsilon$-MOMA) algorithm? If not, it would be very helpful to have a citation to the original reference, since there is not enough detail given here. At the least, the authors should justify how their algorithm differs from Hadka and Reed, and others.

**[Response]** The $\varepsilon$-MOMA is a new MOEA and firstly applied to solve many-objective optimization problems. As stated in the response to specific **Comment #10** above, we have added the process of the algorithm step by step (**Lines 250-288**). The basic framework of $\varepsilon$-MOMA is similar to the traditional NSGAII with significant change in recombination operators and $\varepsilon$-dominance archive from Borg and a local search operator. Borg includes an adaptive population sizing operator that is not used in the proposed algorithm. The strategy adapts the population size in terms of archive size which is considered as a metric of complexity of problems. However, population size will be dramatically increased along with augmentation of archive size, which probably results in a large number of function evaluations. In simulation-optimization method, for CPU-intensive simulation model, this strategy may lead to unaffordable computational burden.

12. Line 253: The discussion of the Ecological Water Conveyance Project is interesting. I'd like to see it integrated better within the text. Is this study supporting that analysis?

**[Response]** The point is well taken. In northwest China, Tarim River, the longest inland river in China, is a typical meandering river that sustains the fragile ecosystem in the basin. However, in the past decades, many tributaries of Tarim River have lost the surface hydraulic interaction with the main stream due to sharply increased water demands. Therefore, Tarim River basin has undergone serious ecological degradation (*e.g.*, land desertification) especially in the lower reaches of Tarim River. In order to restore "Green Corridor" in the lower reaches of Tarim River, Chinese government has implemented the water-conveyance project since 2000 to increase the recharge of groundwater system and raise the local groundwater levels. The project transferred water from Bosten Lake to the Daxihaizi Reservoir and then to the lower reaches of the Tarim River, and finally to the terminal lake (Chen et al., 2010; Yao et al., 2018). Considering the reviewer's concerns, we have added some necessary explanations in the revised text (**Lines 351-356**) and illustrated the details of the water-conveyance project in the **Fig. S1** of Supplementary Materials.

Chen, Y., Chen, Y., Xu, C., Ye, Z., Li, Z., Zhu, C., Ma, X.: Effects of ecological water conveyance on groundwater dynamics and riparian vegetation in the lower reaches of Tarim River, China, Hydrol. Process., 24, 170-177, https://doi.org/10.1002/hyp.7429, 2010.

Yao, J., Chen, Y., Zhao, Y., and Yu, X.: Hydro climatic changes of Lake Bosten in Northwest China during the last decades, Sci. Rep., 8, 9118, https://doi.org/10.1038/s41598-018-27466-2, 2018.

13. Equations 3-5: Why were different metrics used for different variables?

[**Response**] The Nash-Sutcliffe Efficiency (NSE) criterion is a popular method to evaluate model efficiency when the state variables change over time as showed in **Fig. S2a-S2c**. Root mean square error (RMSE) and correlation coefficient (R) are generally used to show goodness-of-fit of calculated and observed variables over the entire stress period as shown in **Fig. S2d**.

14. Line 322-323: This statement should be justified. It speaks to the wider question of how the hydrological modeling is serving the ultimate goal of the management problem, as well as the general contribution of the paper itself. If the focus of the paper is too diffused, it becomes hard to follow its details.

[**Response**] In this study, we firstly built simulation model to evaluate the effect of water management practices on the water resources system. As stated in the revised manuscript (**Lines 433-439**), the water balance of Bosten Lake was calculated by the well-calibrated model and then we found the significance of surface runoff inflow to lake. Therefore, the surface runoff can be considered as the management objective. The analysis paves the way for construction of the management model. Meanwhile, the contribution of the inflow from Kaidu River to Bosten Lake is very close to the result from the previous work of Guo et al. (2015) and Yao et al. (2018).

Guo, M., Wu, W., Zhou, X., Chen, Y., and Li, J.: Investigate of the dramatic changes in lake level of the Bosten Lake in northwestern China, Theor Appl Climatol, 119, 341-351, https://doi.org/10.1007/s00704-014-1126-y, 2015.

Yao, J., Chen, Y., Zhao, Y., and Yu, X.: Hydro climatic changes of Lake Bosten in Northwest China during the last decades, Sci. Rep., 8, 9118, https://doi.org/10.1038/s41598-018-27466-2, 2018.

15. Line 348: To what extent can the groundwater extraction rate be controlled? In some systems, farmers have juristiction on how much to pump. If there is an implicit assumption about a set of water managers who can dictate water usage, this should be stated.

**[Response]** The purpose of the study is to provide suggestions for water managers in local water resources authority. Indeed, some schemes in the Pareto-optimal solutions may be unfeasible for the stakeholders due to the greater extent of regulation of the existing water management scheme. However, a significant advantage of multi-objective optimization is to provide diverse and alternative schemes. The water managers can select the suitable scheme among the Pareto-optimal solutions in terms of specific demands for water management practices. In the optimization, the range of decision variables is specified according to the potential of water use in the irrigation districts or diversion point recorded in the reports of local water resources authority.

16. Lines, 353, 357, etc.: Is there a citation to the water price data? Or was this just an assumption?

**[Response]** The cost coefficients refer to the regulations of the local government. (http://www.xjyq.gov.cn/page.do?danwei=1&fenlei=4000&nian=2017&liushui=19&type=2)

17. Line 406: Guidance on interpreting parallel plots should be provided.

**[Response]** Comment accepted and change made (**Lines 524-526**).

18. Line 534: When the authors say "This study implemented...", were they implying that this occurred across the entire study? Or only in one part of the study? This should be clarified.

**[Response]** Comment accepted. The optimization in three runoff scenarios is the last part of the study to explore the effect of runoff change related to climate change on the water management practices in the basin. We have made some necessary revisions as Dr. Kasprzyk suggested (**Line 651**).

19. Line 541-542: My impression is that hypervolume analyses are usually done to compare optimization runs with the true Pareto set. Is this known? In general, since the optimization

seems to be the focus of this paper, items such as Hypervolume Analysis should be covered in the Methodology (which means that some hydrological modeling detail can be removed)

**[Response]** Comment accepted. For real-world optimization problems, it is computationally expensive to implement many trial runs for the reference Pareto set. In this study, we only calculate the volume of the objective space dominated by a Pareto approximate set (*i.e.*, HV$_{as}$ defined in **Section 2.2.2**). The hypervolume indicator in the section is used to evaluate the optimality of Pareto solutions under different runoff scenarios rather than the convergence and diversity of our proposed algorithm. We have included the hypervolume analysis (**Lines 300-311**) based benchmark test and deleted the statements in the section (**Lines 663-666**).

20. Line 546: "obvious" is usually not appropriate in technical writing.

**[Response]** Comment accepted. We have modified "obviously" as "clearly" (**Line 668)**.

21. Conclusion section: The quality of writing here is much better than in the introduction. Some of this material should inform the Introduction, since this more clearly articulates the purpose of the study than the beginning of the paper did.

**[Response]** Comment accepted. We appreciate the reviewer's positive comment. We have added necessary statements in the Introduction (**Lines 55-57, Lines 176-178,** and **Lines 196-202**).

22. In spite of comment #21, I would like to see slightly more discussion about the management implications of this study - in the local case study as well as how the results can be transferred to other basins (especially given different legal and regulatory structures).

**[Response]** Comment accepted. We have added more discussion in the section "Conclusion" to present more implications (**Lines 724-729**). And the findings are also applicable to regional water resources management in other typical arid inland basins with complex groundwater-river-lake interactions and intensive agricultural development (**Lines 735-737**). As for different legal and regulatory structures for the other basin-scale water management, we need to reconstruct the management model and develop the interactive optimization framework.

23. Table 1: Was random seed analysis performed? If so, the parameters of this analysis should also be provided here. The epsilon values seem quite small - were larger epsilons attempted?

**[Response]** In this study, we didn't perform random seed trials and used the default setting in MATLAB for the rand number generation in which random seed is zero. We performed some optimization trials to select the epsilon value and results show the value in **Table 1** is a good choice. Also, increasing the epsilon value probably reduces the diversity of Pareto solutions.

24. Figure 4: If the paper is too long, I could imagine this figure could go into supplemental material. Also, I noticed that the NSE values appear in the figure but were not referenced in the text.

**[Response]** Comment accepted. We have presented **Fig. 4** of the original manuscript in the supplemental file (**Fig. S2**). The NSE values have been added in the manuscript (**Line 416**; **Line 432**).

25. Figure 9: If possible, the other solutions that are Pareto optimal in 4 dimensions but not in two, should be shown on this plot. Otherwise, the idea that the highlighted solutions fall "outside the front" will be confusing to readers. The Kollat and Reed (2007) paper referenced in this manuscript shows how to do this.

**[Response]** Comment accepted and change made (see **Fig. 8** in the revision).

26. Figure 12: The figure would be easier to understand if the authors reminded reader what these scenarios represent (see also comment #18 - the scenario analysis could be better explained overall).

**[Response]** Comment accepted and change made (**Caption of Fig. 11**).

**[Response]** The proposed new MOEA (epsilon multi-objective memetic algorithm, $\varepsilon$-MOMA) is similar with the algorithmic structure of NSGAII with significant change in the auto-adaptive recombination operator, $\varepsilon$-dominance archive process (Laumanns et al., 2002; Hadka and Reed, 2013) and a local search operator. Comparing with Borg MOEA, $\varepsilon$-MOMA has no change in population size in terms of archive size that is considered as a metric of complexity of problems. The adaptive population sizing probably dramatically increases the number of function evaluations in the optimization, which means the unaffordable computational burden with CPU-intensive model running. Moreover, $\varepsilon$-MOMA revives a local search operator in every several generations of evolutionary search to enhance the local optimality of archived Pareto solutions, which conforms the hybrid framework of MOEA, *i.e.*, multi-objective memetic algorithm (Sindhya et al. 2013).

In the many-objective optimization, the convergence and diversity of Pareto-optimal front are the critical metrics to evaluate the availability of MOEA. The novelty of $\varepsilon$-MOMA is to utilize the $\varepsilon$-dominance concept to archive elite individuals for the maintenance of diversity and the auto-adaptive recombination operator with local search for the enhancement of convergence on the framework of NSGAII. In addition, we linked the proposed MOEA with the numerical model to implement the basin-scale SW-GW management considering complex groundwater-river-lake interactions. To our knowledge, there are no other MOEAs including aforementioned techniques in solving the integrated SW-GW management problems. The detailed algorithmic structure can be found in **Section 2.2.1**.

As stated in the Introduction Section of revised manuscript (**Lines 113-116**), we have firstly stated the difficulty in the many-objective optimization (*i.e.*, the domination-resistance phenomenon). Then we presented two kinds of state-of-art MOEAs by which an attempt to alleviate the difficulty is feasible (**Lines 116-126**). After that, we stated that the developed $\varepsilon$-MOMA attempts to guarantee the diversity and convergence of Pareto-optimal solutions simultaneously in the many-objective optimization. Moreover, this study implemented the benchmark test using the classical DTLZ problems (3-6 objectives) to prove the availability of $\varepsilon$-MOMA (**Table S1** in the Supplementary Materials). Based on above all, we believe that the novelty of the simulation-optimization framework for the conjunctive management of SW and GW developed in this paper deserves consideration for publication in this journal (see the response to the referee's Comment 3 below).

Hadka, D., and Reed, P.M.: Borg: an auto-adaptive many-objective framework, Evol. Comput, 21(2), 213-259, https://doi.org/10.1162/EVCO_a_00075, 2013.

Laumanns, M., Thiele, L., Deb, K., Zitzler, E.: Combining convergence and diversity in evolutionary multi-objective optimization. Evol. Comput, 10(3): 263-282. https://doi.org/10.1162/106365602760234108, 2002.

Sindhya, K., Miettinen, K., Deb, K.: A hybrid framework for evolutionary multi-objective optimization, IEEE Trans., 17(4), 485-511, https://doi.org/10.1109/TEVC.2012.2204403, 2013.

2. You mentioned SFR2, LAK3 and MODFLOW-NWT. Since this is an important part of the framework, can you add more details about these simulation models in the revised manuscript? For example, SFR2 is a streamflow-routing package. Does this model include hydrological simulation, or just a hydraulic model since it is only named as a routing model? How was MODLOW-NWT developed in the study area?

**[Response]** Comment accepted. MODFLOW-NWT is a Newton-Raphson formulation for MODFLOW-2005, which has an obvious advantage in solving drying and rewetting nonlinearities of the unconfined groundwater flow equation (Harbaugh, 2005; Niswonger, et al., 2011). Therefore, MODFLOW-NWT is a newer version of MODFLOW-2005. Most modular packages supported by MODFLOW-2005 can be used with MODFLOW-NWT, including SFR2 and LAK3 packages.

Streamflow-Routing Package (SFR2), as a modular package in MODLFOW-NWT, can be used to model the interactions between streams and underlying aquifers while consider unsaturated flow beneath streams for the disconnected river (Richard and David, 2010). SFR2 is just a modular package to model streamflow in the river channel based on the continuity equation assuming steady and uniform flow rather than an independent hydrological simulation model. The Manning's Equation is used to represent the relation between river stage and discharge and Darcy's Law is used to calculate the infiltration/exfiltration rate between streams and aquifers.

Considering the referee's concerns, we have added more details of the modular packages to elucidate the model (**Lines 372-375, Lines 382-386, Lines 388-394**).

The model used SFR2 package to simulate the streamflow routing in Kaidu River and surface water diversion to the 11 aqueducts from the mainstream of Kaidu River. The LAK3 package was used to model the variation of lake level of Bosten Lake in response to lake atmospheric recharge evaporation, surface runoff inflow from Kaidu River and withdrawal of ecological water conveyance. The runoff in gaging stations and observed lake levels were used to calibrate the parameters of SFR2 and LAK3. The observed groundwater levels were employed to calibrate the regional groundwater flow process.

Richard, G.N. and David, E.P.: Documentation of the Streamflow-Routing (SFR2) Package to Include Unsaturated Flow Beneath Streams-A Modification to SFR1, U.S. Geological Survey Techniques and Methods, pp. 6-A13, 2010.

Michael, L.M. and Leonard, F.K.: Documentation of a Computer Program to Simulate Lake-aquifer Interaction Using the Modflow Ground-water Flow Model and the Moc3d Solute-transport Model, U.S. Geological Water-Resources Investigations Report, 2000.

Harbaugh, A.W.: MODFLOW-2005, the U.S. Geological Survey modular ground-water model - the Ground-Water Flow Process: U.S. Geological Survey Techniques and Methods 6-A16, 2005.

Niswonger, R.G., Panday, S., and Ibaraki, M.: MODFLOW-NWT, A Newton formulation for MODFLOW-2005: U.S. Geological Survey Techniques and Methods 6-A37, 44 p, 2011.

3. Was the epsilon MOMA algorithm developed by yourself? The references attached are not enough to understand the algorithm. Please add more details about the logic line of the algorithm.

[**Response**] Yes. As stated in **Comment #1**, we developed $\varepsilon$-MOMA algorithm based the previous work of the state-of-the-art MOEAs. **In Section 2.2.1**, we have presented the logic line of the proposed algorithm step by step (**Lines 250-288**).

4. Figure 2: the figure didn't show clearly the river names in the basin. For example, I cannot find Kongqi River and lower Tarim River in the figure. This figure should help us understand the rivers, aqueducts etc. Please add a more detailed map.

**[Response]** Comment accepted. In the revised manuscript, **Fig. 2** has been modified to present all the rivers (*i.e.*, Kaidu River, Huangshuigou River, Qingshui River, Kongqi River) and 11 main aqueducts in the study area. To state the artificial outflow of Bosten Lake, we have also briefly introduced the Ecological Water Conveyance Project in the revised manuscript (**Lines 351-356**), although the project is not focus of our study. Considering the referee's concerns, we have illustrated the project in the **Fig. S1** of Supplementary Materials to present all the rivers stated in the manuscript.

5. When setting up the simulation model, what kind of data and also the details of data should be explained. What data were used for model calibration and validation?

**[Response]** Considering the referee's concerns, we have clearly listed the data for model set-up in the **Table S2** of Supplementary Materials (**Lines 400-401**). **Table S2** provides the data details used to build up the model in this study which can be grouped into three categories. The first category is the data depicting hydrological features and stratigraphic characteristics, including the spatial structure of regional aquifer and Bosten Lake, the network of Kaidu River and aqueducts. The second category is to input dynamic source and sink terms including boundary groundwater flux and groundwater level, weather observations, boundary river inflow, artificial pumping from Bosten Lake and conjunctive use of surface water and groundwater for agricultural irrigation. The third category is the hydrological observation data for model calibration. Due to the data scarcity, all available observation data is used to calibrate the numerical model.

6. What is "stress period"?

**[Response]** The simulation period is divided into a series of "stress period" within which specified stress data are constant. The stress data include the finite-difference cell dimension, time information, boundary conditions, initial heads, aquifer hydraulic properties, and control information required by the numerical solution scheme. The concept of stress period is the fundamental components in solving groundwater flow process using MODFLOW program (Harbaugh, 2005), which is similar to "time step" in hydrologic model.

Harbaugh, A.W., 2005, MODFLOW-2005, the U.S. Geological Survey modular ground-water model -- the Ground-Water Flow Process: U.S. Geological Survey Techniques and Methods 6-A16.

7. What is the time resolution in your simulation model? From Fig.4, you can see that the resolution is very coarse, semiannually? This model fails to show even the seasonality of runoff, lake level and water allocation.

[Response] As stated in the manuscript (Lines 394-397), the length of stress period in the non-irrigation and irrigation period is 5 and 7 months, respectively. We acknowledge that the time resolution is relatively coarse in the hydrological modeling subject to the availability of the data of groundwater abstraction and surface water diversion in the monthly scale, and lack of long-term monitoring of groundwater level during simulation period. However, Yanqi Basin is an arid inland basin with intensive agricultural development. Therefore, agricultural irrigation dominates the dynamic of groundwater level due to groundwater abstraction, and the variation of streamflow in Kaidu River due to surface water diversion in the basin. From the perspective of water managers, the model reflects the key relations between water resources exploitation and agricultural development, that's the critical components of multi-objective simulation-optimization modeling. Therefore, the model can be used to implement the integrated management of surface water and groundwater in Yanqi Basin.

8. How did the simulation consider all human activities in the model? For example, how SFR2 take into account the diversion or abstract of water from the river?

[Response] In Yanqi Basin, human activities on water resources include groundwater abstraction and surface water diversion. Firstly, we have the spatial location and yearly pumping rate of all pumping wells only in 2009 from national water resources census. The groundwater abstraction is mainly used for the crop water demand in the basin. Since we only have yearly total pumping rates from 2003 to 2013 in the basin, we reallocate the total pumping rates into each well in the other years according to the percentage of each pumping rate calculated in 2009. Then we disaggregated the yearly pumping data in each well into non-irrigation and irrigation period based on the investigation of local farmers' irrigation behaviors. Finally, the pumping data are rescaled to temporal and spatial scales required by the numerical model. The Well Package, as a modular package in MODFLOW-NWT, is used to model groundwater abstraction.

In SFR2 package, we can specify the water volume of surface water diversion in the diversion point of each aqueduct, which are recorded in the reports from the local water resources authority. Moreover, the aqueduct can be considered a "river" to model the water exchange with the underlying aquifer using SFR2 package. In the optimization, the amount of surface water diversion in seven diversion points can be considered as the decision variables to implement optimal surface water allocation in the basin.

9. Page 17: How did you obtain the scheme before optimization?

**[Response]** As stated in the manuscript (**Lines 499-501**), the management period is the last two stress periods (from November 2012 to October 2013) of the simulation period (from November 2003 to October 2013). Therefore, the scheme before optimization is known.

10. Climate change has substantial impacts on river runoffs in arid rivers in Xinjiang Province. The authors used only three simple scenarios (Current runoff; reduce 10% runoff and reduce 20% runoff) to investigate the impacts of climate change. These scenarios are just toys and don't provide useful information for climate change adaptation for the study area. Why didn't the authors use more practical climate change scenarios like RCPs?

**[Response]** The point is well taken. The climate-driven changes have a significant effect on the streamflow in snow-fed and glacier-fed basins, *e.g.*, Kaidu River Basin (KRB). The KRB drains an area of 18649 km$^2$ above Dashankou (DSK) gauge station and is considered as the upper mountainous headwater regions of Yanqi Basin (Shen, et al., 2018). The streamflow in DSK station, dominated by the hydrological regimes in KRB, is of critical importance for the integrated management of surface water and groundwater in Yanqi Basin. Ba et al. (2018) employed the SWAT model with three RCMs (regional climate models) to analyze the influences of climate change on the streamflow in DSK station. The study results show that the annual streamflow will decreases during 2020-2049 and reaches to the largest reduction percentage of 20.1% and 22.3% during 2040-2049 under RCP4.5 and RCP8.5 scenarios, respectively. Therefore, in this study, we defined the three runoff scenarios (*i.e.*, current runoff, reduce 10% runoff and reduce 20% runoff) based the work of Ba et al., (2018) to elucidate the impacts of climate change. However, the much heterogeneity in the climate change models and assumptions results in the uncertainty of runoff prediction in the future climate scenarios, that's not the focus of our study. Considering the referee's concerns, we have added the Reference and modified the statement in the manuscript (**Lines 653-660**).

Ba, W., Du, P., Liu, T., Bao, A., Luo, M., Mujtaba, H., and Qin, C.: Simulating hydrological responses to climate change using dynamic and statistical downscaling methods: a case study in the Kaidu River Basin, Xinjiang, China. J. Arid Land, 10(6): 905-920, https://doi.org/10.1007/s40333-018-0068-0, 2018.

Shen, Y.J., Shen, Y., Fink, M., Kralisch, S., and Brenning, A.: Unraveling the hydrology of the glacierized Kaidu Basin by integrating multisource data in the Tianshan Mountains, Northwestern China, Water Resour. Res., 54: 557-580. https://doi.org/10.1002/2017WR021806, 2018.

11. Possible uncertainty in the simulation-optimization model and decision making should be discussed in the manuscript.

**[Response]** Indeed, we acknowledge that the parameter uncertainty with limited data availability in the basin-scale full-coupled model and the limitations of model structure are inevitable. However, in this study, the numerical model can reflect the responses of water resources system to the conjunctive use of surface water and groundwater for agricultural irrigation. The simulation-optimization model also provides insights into basin-scale water resources management in the Yanqi Basin or other arid land basins with intensive agricultural development. The parameter uncertainty can be addressed with the construction of adequate and sustainable observing system in the future work. Considering the referee's concerns, we have modified the statements in the revised manuscript (**Lines 735-741**). In addition, the study focuses on the many-objective optimization of water allocation in the basin under the certain environment. The possible uncertainty (*e.g.*, deep uncertainty) also can be considered to implement many-objective robust optimization under the noisy environment (Watson and Kasprzyk, 2017), that's the focus of our future work (**Lines 744-746**).

Watson, A.A., and Kasprzyk, J.R.: Incorporating deeply uncertain factors into the many objective search process, Environ. Model. Softw., 89: 159-171. http://dx.doi.org/10.1016/j.envsoft.2016.12.001, 2017.

Technical corrections: Only minor typo is found.

**[Response]** Comment accepted. We have thoroughly checked the manuscript and revised the original manuscript to improve its quality and readability.

**Response to Dr. Qiankun Luo's Short Comments**

This manuscript focus on the topic of conjunctive using of surface water and groundwater based on the multi-objective simulation and optimization method. In the manuscript, a novel multi-objective optimization model with four objective functions is developed to balance the water demand for agriculture, socioeconomic development and environmental demands. In order to find the Pareto optimal solutions for the special model, a new multi-objective evolutionary algorithm, named $\varepsilon$-MOMA, is presented. The optimization results of Yanqi Basin (YB) in northwest China certified the applicability of the new model and optimization algorithm. Generally, the manuscript does make an important contribution on water resources management research. However, there are some general and specific comments referencing lines in the manuscript which will be helpful for the improvement of the manuscript.

**[Response]** We appreciate Dr. Qiankun Luo's insightful comments and constructive suggestions. We have fully addressed the concerns into the revised manuscript and given a point-by-point response as below.

General comments:

1. The introduction usually includes the research background, the research problems, a review of the advantages and disadvantages of the previous and latest research results, and the new solving method of the present research. Thus, the description of the detailed condition of the study area should be move into section 3.1.

**[Response]** Considering the referee Dr Luo's concerns, we have reorganized the statements in the Introduction. Firstly, we clarified the necessity of water resources management in the arid inland basin to present the motivation of our study (**Lines 53-63**). Secondly, we explained the meaning of many-objective optimization framework in the water resources management and planning (**Lines 81-109**) and the optimization techniques (**Lines 110-137**). After that, we introduced the details of water resources exploitation in Yanqi Basin (YB) (**Lines 144-160**) to present the suitability of the case study. Then, we stated that it is necessary for YB water management to consider the deep uncertainty derived from climate change which probably results in the runoff reduction in Kaidu River (**Lines 182-190**). Finally, we showed the general results of the study for the HESS readers (**Lines 196-202**).

2. The advantage of the newly developed optimization algorithm should be given in detail. For example, why the $\varepsilon$-MOMA is better than the other MOEA in solving groundwater management problems? What is the main difference between $\varepsilon$-box and the elite individual preservation strategy?

**[Response]** The point is well taken. We have split **Section 2.2** into **Section 2.2.1** "Main algorithmic structure" to present the process of the new algorithm step by step (**Lines 250-288**) and **Section 2.2.2** "Benchmark test" to investigate the performance of the algorithm (**Lines 296-313**). The new algorithm $\varepsilon$-MOMA, which used several promising techniques from Borg MOEA and a local search operator to improve the optimality of Pareto solutions, has been validated to present the effectiveness in solving the many-objective optimization. In this study, it is not the focus to implement comparative study of the state-of-the-art MOEAs in solving water resources management problems. The benchmark tests also show the availability of our algorithm (**Table S1** in the Supplementary Materials). After all, our study aims to propose a promising multi-objective optimization framework for the integrated surface water and groundwater management in the typical arid inland basin.

The concepts of Pareto dominance and $\varepsilon$-dominance can be defined as follows and we assume that all objectives are to be minimized. The vectors $\boldsymbol{f}=(f_1, f_2, f_3,\ldots, f_m)$ and $\boldsymbol{g}=(g_1, g_2, g_3,\ldots, g_m)$ can be denoted as objective values where $m$ is the number of objectives and $\boldsymbol{\varepsilon}=(\varepsilon_1, \varepsilon_2, \varepsilon_3,\ldots,\varepsilon_m)$ is the allowable tolerance vector specified by the users.

The objective vector $\boldsymbol{f}$ is said to Pareto dominate $\boldsymbol{g}$, if:

$$\begin{aligned} f_i \le g_i &\quad \forall i \in \{1,...,m\} \\ f_i < g_i &\quad \exists i \in \{1,...,m\} \end{aligned} \tag{1}$$

The objective vector $\boldsymbol{f}$ is said to $\varepsilon$-dominate $\boldsymbol{g}$, if:

$$\left(1-\varepsilon_i\right) f_i \le g_i \quad \forall i \in \{1,...,m\} \tag{2}$$

The $\varepsilon$-dominance allows the decision-makers to specify the resolution of the Pareto set approximation by selecting an appropriate $\varepsilon$ value while guarantees the diversity of Pareto solutions over the optimal Pareto front (Laumanns et al., 2002; Deb et al., 2005).

Laumanns, M., Thiele, L., Deb, K., and Zitzler, E. (2002). Combining convergence and diversity in evolutionary multi-objective optimization. Evolutionary Computation, 10(3): 263-282. https://doi.org/10.1162/106365602760234108.

Deb, K., Mohan, M., and Mishra, S. (2005). Evaluating the ε-domination based multi-objective evolutionary algorithm for a quick computation of Pareto-optimal solutions. Evolutionary Computation, 13(4): 501-525. https://doi.org/10.1162/106365605774666895.

3. In the numerical simulation process, the irrigation backflow should be considered. How to deal with the irrigation backflow in the groundwater flow numerical simulation model of the YB?

**[Response]** The point is well taken. The irrigation water including the surface water (SW) in an aqueduct system and the groundwater (GW) in the regional aquifer. We allocate SW

diverted from Kaidu River to an irrigation district according to the source of irrigation water derived from which aqueduct. The GW can be allocated in terms of the locations of pumping wells distributed in the irrigation district. The return flow from agriculture irrigation can be calculated by multiplying the irrigation water demands by the irrigation infiltration coefficient based on reports from the local water resources authority.

4. YB is a typical arid inland basin in China. The optimization results of YB are seemed reliably, can this optimization model be used directly in other basin or field?

**[Response]** The proposed many-objective optimization framework can be extended to solve the integrated SW-GW management problems (**Lines 735-737**) once the simulation model can be built in the other basins or fields. The simulation model can be developed with the fully-coupled hydrological model to reduce the prediction error derived from numerical model that is our focus in the future study.

Specific comments, where line numbers refer to the PDF version of the HESSD paper:

1. Line 168: I suggest to changing the "decision-maker" to "water manager" in the manuscript. The author sometimes uses "decision-maker" and sometimes uses "water manager", which will confuse the readers.

**[Response]** Comment accepted and change made (**Line 231**). We have modified the "decision-maker" to "water manager" in the context of elucidating water resource management throughout the manuscript.

2. Line 199-201: Did all of the referred recombination operators (SBX, DE, SPX, PCX, LX, UM) used in the new optimization method? Or only one of them was adopted? The author should clear it.

**[Response]** As stated in the manuscript (**Lines 258-267**), the crossover probability of each recombination operator is updated periodically based on the proportion of the solutions generated by the operator in the $\varepsilon$-dominance archive. In the optimization, we firstly assign same probability for all of the operators which can be used in the preliminary stage. The optimal operator can be chosen with the highest probability at the later stage of evolutionary search.

3. Line 291: There is a mistake in Equation 4, the "2" was lost.

**[Response]** Comment accepted and change made (**Line 408**).

4. Line 417: where the increment of $f_{TPR}$ and $f_{TDR}$ from, the explanation should be given.

**[Response]** The increments indicate the range of $f_{TPR}$ and $f_{TDR}$ across all the Pareto solutions to show the extent of regulation of groundwater abstraction and surface water diversion in the post-optimization (**Lines 536-537**).

5. Line 539-541: Why the lake level is changed to a smaller value? And why the maximum groundwater drawdown is reset to 10m?

**[Response]** The reduction of runoff in Kaidu River directly lowers the runoff inflow to the terminal lake which results in the decline of lake level. Meanwhile, the groundwater exploitation must be augmented to offset the reduction of available runoff for irrigation water demands, which increases the groundwater drawdown in the regional aquifer. In the optimization, the constraints of minimum lake level and maximum groundwater drawdown need to be altered to avoid much more infeasible solutions in the population which inhibits convergence of the MOEA. The optimization under Scenarios A0, A1 and A2 is to implement comparative analysis for quantifying the effect of runoff reduction on the YB water management.

6. Line 557: "a certain value" should be given explicitly for the case study.

**[Response]** The point is well taken. We have modified the statement in the revised manuscript (**Lines 678-681**).

7. Line 578: Change "Yanqi Basin" to "YB".

**[Response]** Comment accepted (**Line 701**).

[revised manuscript text omitted]

**Figures**

[Figure]

**Fig. 1.** Framework of multi-objective optimization for integrated SW-GW management.

[Figure]

**Fig. 2.** The location of Yanqi Basin and the model domain of interest for this study. Source:

DigitalGlobal, Inc. (imagery).

[Figure]

**Fig. 3.** The boundary conditions of model domain, monitoring locations of groundwater level and surface runoff, aqueduct system and bathymetric contours in meters for Bosten

Lake.

[Figure]

**Fig. 4.** The water balance terms of Bosten Lake and resulting lake volume in the simulation
period.

[Figure]

**Fig. 5.** The locations of surface water diversion points and subdomains of irrigation districts for

 groundwater abstraction.

[Figure]

**Fig. 6.** The tradeoff surface to the integrated SW-GW management in Yanqi Basin. Each spheric symbol represents a water use scheme corresponding to specific objective values of the total water supply rate ($f_{TWS}$), total cost of water delivery ($f_{TCOST}$), surface runoff inflow to lake ($f_{SRI}$) and groundwater storage change ($f_{GSC}$). $f_{TCOST}$ is symbolized in color to identify the objective value against others. The green arrow is the direction of better performance for each objective. The scheme before optimization is marked in a red square box.

[Figure]

**Fig. 7.** The objective values (*y*-axis) are plotted over management objectives $f_{TWS}$, $f_{TCOST}$, $f_{GSC}$, $f_{SRI}$, total pumping rate $f_{TPR}$ and total surface water diversion rate $f_{TDR}$ (*x*-axis), $f_{TWS}$ is represented in color. The preferred direction for each index is upward.

[Figure]

**Fig. 8.** Identification of six interesting solutions (Solutions 1-6) from the four-dimensional
approximate Pareto set and the green arrow is the preferred direction for each objective.

[Figure]

**Fig. 9.** The spatial distribution of the pumping rates in the 11 irrigation districts for the four selected schemes of (a) Solution 4, (b) Solution 7, (c) Solution 5, and (d) Solution 6, respectively.

[Figure]

**Fig. 10.** Variation of surface runoff and river leakage along the stem stream of Kaidu River: (a)

the profile of river runoff; (b) the distribution of surface water diversion at the different diversion points; (c) the profile of river leakage; (d) the components of total river leakage, groundwater abstraction and surface water diversion for several typical Solutions 4-7.

[Figure]

**Fig. 11.** The tradeoff solutions under Scenarios A0 (maintain current runoff), A1 (reduce the runoff by 10%) and A2 (reduce the runoff by 20%), and the sphere size indicates the value of $f_{TCOST}$. The green arrow is the direction of better performance for each objective.

[Figure]

**Fig. 12.** Evolution of the hypervolume metric over the generation number for Scenarios A0, A1

 and A2.

[Figure]

**Fig. 13.** Non-dominated fronts of Scenarios A0, A1 and A2 between objectives of $f_{GSC}$ vs. $f_{SRI}$.

*Supplementary Materials for*

**Basin-scale multi-objective simulation-optimization modeling for**

**conjunctive use of surface water and groundwater in northwest China**

Jian Song[a], Yun Yang[b], Xiaomin Sun[c], Jin Lin[c], Ming Wu[d], Jianfeng Wu[a,*], Jichun Wu[a]

[a] Key Laboratory of Surficial Geochemistry, Ministry of Education; Department of
  Hydrosciences, School of Earth Sciences and Engineering, Nanjing University, Nanjing,
  210023, China

[b] School of Earth Sciences and Engineering, Hohai University, Nanjing, 210098, China

[c] Nanjing Hydraulic Research Institute, National Key Laboratory of Water Resources and
  Hydraulic Engineering, Nanjing, 210029, China

[d] Institute of Groundwater and Earth Sciences, Jinan University, Guangzhou, 510632, China

*Correspondence to*: Jianfeng Wu (jfwu@nju.edu.cn; jfwu.nju@gmail.com)

**Table S1** The control parameters and hypervolume metric obtained for $\varepsilon$-MOMA on

*M*-objective DTLZ1 and DTLZ3 problems

| Problem | $M$ | $N_{dv}$ | $N_{pop}$ | $N_{eval}$ | $\varepsilon_{obj}$ | $rp$ | $HV_{rs}$ | $HV_{as}$ | $HV_n$ |
|---|---|---|---|---|---|---|---|---|---|
| | 3 | | | 100,000 | | | 0.14575 | 0.14480 | 0.9935 |
| | 4 | | | 150,000 | | | 0.08883 | 0.08828 | 0.9939 |
| DTLZ1 | | $M$+9 | 200 | | 0.01 | 0.55 | | | |
| | 5 | | | 200,000 | | | 0.05000 | 0.04982 | 0.9964 |
| | 6 | | | 400,000 | | | 0.02763 | 0.02759 | 0.9985 |
| | 3 | | | 100,000 | | | 0.63507 | 0.61857 | 0.9740 |
| | 4 | | | 150,000 | | | 0.89568 | 0.85577 | 0.9554 |
| DTLZ3 | | $M$+9 | 200 | | 0.01 | 1.05 | | | |
| | 5 | | | 200,000 | | | 1.08860 | 1.03550 | 0.9512 |
| | 6 | | | 400,000 | | | 1.23140 | 1.19210 | 0.9681 |

Note: $M$ = number of objectives; $N_{dv}$ = number of decision variables; $N_{pop}$ = population size;

$N_{eval}$ = number of function evaluations; $\varepsilon_{obj}$ = epsilon value for each objective; $rp$=the value of reference point for each objective; $HV_{rs}$ = hypervolume of Pareto reference set; $HV_{as}$ =

hypervolume of Pareto approximate set; $HV_n$ = the normalized hypervolume.

**Table S2** Multisource data for the model build-up

| Category | Data | Data Time | Spatial Resolution |
|---|---|---|---|
| Initial parameterization and resolution | DEM | 2008 | 90×90 m |
| | River network | 2009 | (Google Map) |
| | Aqueducts | 2009 | (Reports) |
| | Hydrogeology Map | 1977 | 1:200000 |
| | Lake topography | 1977 | 1:200000 |
| | Bottom of aquifer | 1977 | 1:200000 |
| Dynamic data and resolution | Boundary river inflow | 2003-2012 (monthly) | 1 station |
| | Boundary groundwater inflow | 2009 (yearly) | (Reports) |
| | Boundary groundwater level | 2003-2013 (non-irrigation and irrigation periods) | 5 monitoring wells |
| | Meteorological observations | 2003-2013 (monthly) | 3 stations |
| | Surface water diversion | 2003-2013 (non-irrigation and irrigation periods) | 11 aqueducts |
| | Groundwater pumping | 2003-2013 (yearly) | 11 irrigation districts |
| | Lake artificial pumping | 2003-2013 (monthly) | 1 station |
| Calibrated data and resolution | Streamflow | 2003-2012 (monthly) | 2 stations |
| | Groundwater level | 2003-2013 (non-irrigation and irrigation periods) | 7 wells (2003-2013) 14 wells (2012-2013) |
| | Lake level | 2003-2013 (monthly) | 1 station |

[Figure]

**Fig. S1** The Ecological Water Conveyance Project

[Figure]

**Fig. S2** The calibrated results of the transient model showing (a) observed vs. calibrated runoff at Yanqi station over time, (b) observed vs. calibrated runoff at Baolangsumu station over time; (c) observed vs. calibrated lake level over time; (d) comparison of observed and calibrated groundwater heads at all observation wells, and (e) observed vs. calibrated groundwater heads over time at three typical observation locations as labeled in Fig. 3.